# NASH: A Simple Unified Framework of Structured Pruning for Accelerating Encoder-Decoder Language Models

**Jongwoo Ko**[1*]   **Seungjoon Park**[2*‡]   **Yujin Kim**[1]   **Sumyeong Ahn**[3†‡]
**Du-Seong Chang**[2]   **Euijai Ahn**[2]   **Se-Young Yun**[1†]
[1]KAIST AI    [2]KT    [3]Michigan State University
https://github.com/jongwooko/NASH-Pruning-Official

## Abstract

Structured pruning methods have proven effective in reducing the model size and accelerating inference speed in various network architectures such as Transformers. Despite the versatility of encoder-decoder models in numerous NLP tasks, the structured pruning methods on such models are relatively less explored compared to encoder-only models. In this study, we investigate the behavior of the structured pruning of the encoder-decoder models in the decoupled pruning perspective of the encoder and decoder component, respectively. Our findings highlight two insights: (1) the number of **decoder layers** is the dominant factor of inference speed, and (2) low sparsity in the pruned **encoder network** enhances generation quality. Motivated by these findings, we propose a simple and effective framework, **NASH**, that narrows the encoder and shortens the decoder networks of encoder-decoder models. Extensive experiments on diverse generation and inference tasks validate the effectiveness of our method in both speedup and output quality.

## 1   Introduction

In recent years, pre-trained language models (LMs) have demonstrated their effectiveness in various downstream tasks, such as natural language understanding (NLU) and natural language generation (NLG). Especially, there have been three main types of research, *e.g.,*encoder-only LMs (Devlin et al., 2019; He et al., 2023), decoder-only LMs (Touvron et al., 2023; OpenAI, 2023), and *encoder-decoder* LMs (Lewis et al., 2020; Raffel et al., 2020; Chung et al., 2022b; Tay et al., 2023), which aim for their specific expertise. Among these various types of LMs, we will focus on the widely studied and utilized *encoder-decoder* LMs due to their flexibility in application across a range of tasks (Guo et al., 2022; Wang et al., 2023b).

---
*denotes equal contribution.    [†] denotes equal advising.
[‡] denotes working done at KAIST AI.    Correspondence to Jongwoo Ko <jongwoo.ko@kaist.ac.kr> .

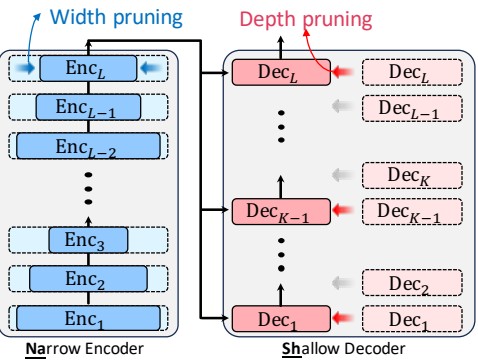

Figure 1: Brief illustration of the proposed algorithm, **NASH**: **NA**rrow encoder and **SH**allow decoder. It is composed of two main components, width and depth pruning for encoder and decoder, respectively.

On the other perspective of LM researches rather than performances, efficiency of LMs (*e.g.,*computational and memory cost) have been intensively studied because of their huge computational requirements. This research direction is called model compression. Among the various model compression techniques (Jiao et al., 2020; Yao et al., 2022), pruning (Frankle and Carbin, 2018; Sanh et al., 2020; Wang et al., 2020c; Xia et al., 2022) is a promising method that aims to remove redundant weights from networks, resulting in improved efficiency by saving storage capacity and enhancing inference speed. Between structured pruning and unstructured pruning approaches, structured pruning is typically preferred in practice due to its relative ease of deployment on various types of hardware platforms compared to unstructured pruning (Han et al., 2016; Gupta and Agrawal, 2020).

Therefore, we focus on the structured pruning method specifically tailored for *encoder-decoder* LMs. Despite the remarkable advancements in *encoder-decoder* models, little attention has been given to structured pruning methods for *encoder-decoder* LMs. This can be attributed to the inherent

differences in the components that enhance pruning efficiency between encoder and decoder networks. Consequently, traditional structured pruning methods that rely on encoder-only models may not effectively optimize *encoder-decoder* models. For instance, CoFi (Xia et al., 2022), one of the SoTA encoder-only pruning methods, demonstrates a maximum speedup improvement of $1.53\times$ on the CNN/DailyMail (See et al., 2017) dataset, with a ROUGE-L drop of $7.36\%$. This gain is considerably lower compared to the original result achieved on the encoder-only model applied to the worst QQP case in the GLUE (Wang et al., 2018), where the speedup reaches $11.0\times$ with an accuracy drop of $1.20\%$. Thus, it becomes crucial to investigate structured pruning methods that are specifically tailored for the encoder and decoder networks.

To this end, in this paper, we pose the following question: How can we design a structured pruning method that effectively accelerates the *encoder-decoder* model while maintaining its performance? To the best of our knowledge, this study represents the first attempt to address this question. In order to accomplish this, we conduct systematic studies to examine the impact of structured pruning on the encoder and decoder networks, respectively.

**Contribution.** In this study, we propose an algorithm, NASH, which is strongly motivated by two findings derived from our preliminary experiments. (1) **The number of decoder layers** is the primary factor for inference speedup. (2) **The sparsity of the encoder network** is a key factor affecting the output quality of *encoder-decoder* LMs.

Based on these findings, we propose an algorithm, illustrated in Figure 1, that consists of two parts: the encoder network, which enhances output quality by gradually reducing the width of each layer, and the decoder network, which achieves faster inference speed by uniformly selecting layers to reduce depth.

We empirically evaluate the performance of NASH on various NLG datasets including standard fine-tuning on a single task (Gliwa et al., 2019; Xiong et al., 2019), multi-task learning scenarios, and recent instruction-tuning datasets (Conover et al., 2023; Wang et al., 2023a). Notably, in our experiments using T5-base, NASH achieves a speedup of 2.5-4.2$\times$ while preserving $95\%$ of the output quality. Our experimental results show that NASH can be a unified framework which is regardless of task difficulty and model type.

## 2 Preliminary

**Transformers.** We focus on the Transformer network (Vaswani et al., 2017), which consists of the encoder and decoder architecture. The encoder architecture is composed of $L$ blocks, and each block consists of a multi-head attention (MHA) layer and a feed-forward (FFN) layer. An MHA layer in the $i$-th Transformer layer with $N_h$ heads outputs:

$$\text{MHA}_{(i,j)}(\mathbf{Q}, \mathbf{K}, \mathbf{V}) = \text{Att}(\mathbf{Q}\mathbf{W}_Q^{(i,j)}, \mathbf{K}\mathbf{W}_K^{(i,j)}, \mathbf{V}\mathbf{W}_V^{(i,j)}),$$

$$\text{MHA}_{(i)}(\mathbf{Q}, \mathbf{K}, \mathbf{V}) = \sum_{j=1}^{N_h} \text{MHA}_{(i,j)}(\mathbf{Q}, \mathbf{K}, \mathbf{V})\mathbf{W}_O^{(i,j)},$$

where Att represents a dot product attention head, and $\mathbf{Q}$, $\mathbf{K}$, and $\mathbf{V}$ are the input sequences for query, key, and value, respectively. In self-attention layers, all of the keys, values, and queries come from the outputs of the previous layer. On the other hand, in cross-attention layers, the queries come from the previous decoder layer, while the memory keys and values come from the output of the encoder. It is important to note that the $j$-th head is parameterized by $\mathbf{W}_Q^{(i,j)}, \mathbf{W}_K^{(i,j)}, \mathbf{W}_V^{(i,j)}$, and $\mathbf{W}_O^{(i,j)} \in \mathbb{R}^{d \times d_h}$, which represent the query, key, value, and output matrices, respectively. Here, $d$ and $d_h$ denote the hidden state dimension and attention head dimension, respectively.

The output of the MHA layer, denoted as $\mathbf{X}$, is then fed into the FFN layer in the $i$-th Transformer layer:

$$\text{FFN}_{(i)}(\mathbf{X}) = \text{GELU}(\mathbf{X}\mathbf{W}_1)\mathbf{W}_2.$$

Here, the two fully-connected layers are parameterized by $\mathbf{W}_1 \in \mathbb{R}^{d \times df}$ and $\mathbf{W}_2 \in \mathbb{R}^{df \times d}$, with $d_f$ representing the dimension of the FFN layer.

**Structured Pruning.** Structured pruning gradually removes unnecessary parameters from a model, targeting width-related components (*e.g.,* MHA heads, FFN hidden units) and depth-related elements (*e.g.,* Transformer layers) during training. Recent advancements have demonstrated significant speedups with minimal output quality reduction. For example, block pruning (Lagunas et al., 2021) and CoFi (Xia et al., 2022) have enhanced flexibility, optimization, and enabled simultaneous pruning at multiple levels.

Pruning the components of the $i$-th layer related

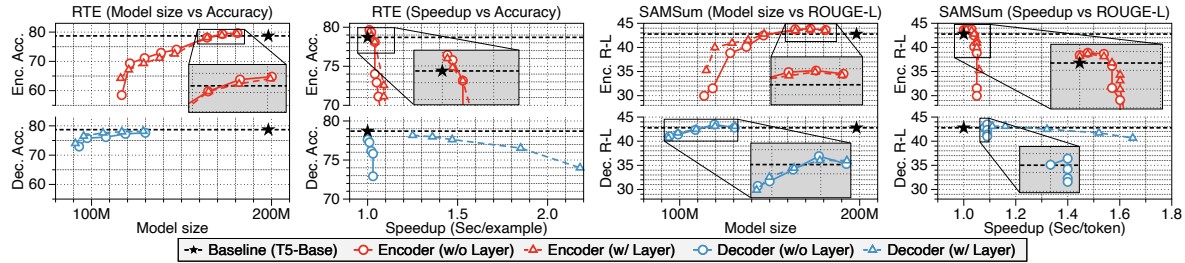

Figure 2: Comparing model size (or speedup) vs. output performance for four pruning options: with or without depth pruning applied to either the encoder or decoder network individually. The results emphasize that (1) **the number of layers in the decoder network** is the primary factor contributing to speedup improvements. and (2) **the sparsity of the encoder network** is the key factor of output quality.

to MHA can be formulated as follows:

$$\text{MHA}_{(i)}(\mathbf{Q}, \mathbf{K}, \mathbf{V}) = z_{\text{MHA}}^{(i)} \cdot \sum_{j=1}^{N_h} \mathbf{z}_{\text{head}}^{(i,j)} \cdot$$
$$\text{Att}(\mathbf{Q}\mathbf{W}_Q^{(i,j)}, \mathbf{K}\mathbf{W}_K^{(i,j)}, \mathbf{V}\mathbf{W}_V^{(i,j)})\mathbf{W}_O^{(i,j)},$$

where $z_{\text{MHA}}^{(i)}$ and $\mathbf{z}_{\text{head}}^{(i,j)} \in \{0, 1\}$ and to mask MHA layer and individual head of MHA.

The FFN layer, which is another major component of the Transformer network, is also known to be over-parameterized (Dong et al., 2021). Strategies for pruning include pruning an entire FFN layer and pruning intermediate dimensions at a more granular width level. This can be achieved by introducing mask variables, $z_{\text{FFN}}^{(i)}$ and $\mathbf{z}_{\text{int}}^{(i)} \in \{0, 1\}^{d_f}$, with the following formulation:

$$\text{FFN}_{(i)}(\mathbf{X}) = z_{\text{FFN}}^{(i)} \cdot \text{GELU}(\mathbf{X}\mathbf{W}_1) \cdot \text{diag}(\mathbf{z}_{\text{int}}^{(i)}) \cdot \mathbf{W}_2$$

Various techniques have been employed to learn these mask variables used in structured pruning. For example, Wang et al. (2020c) and Xia et al. (2022) utilized $L0$ regularization to eliminate redundant parameters. On the other hand, Lagunas et al. (2021) adopted the movement score introduced by Sanh et al. (2020) as a measurement for their pruning approach.

## 3 Experimental Motivations

In this section, we separately investigate the behavior of the encoder and decoder when depth pruning is applied or not, using CoFi-T5, the modified version of CoFi (Xia et al., 2022) tailored for T5 (Raffel et al., 2020). Particularly, in Figure 2, we study the results of four cases: encoder with depth-pruning (△), encoder without depth-pruning (○), decoder with depth-pruning (△), and decoder without depth-pruning (○). From these four types

of cases, we aim to address the following questions: (1) Does depth pruning exhibit different phenomena in each case? (2) What is the key factor for accelerating inference speed while preserving sufficient output quality? We provide detailed answers to each question by training T5-Base with target sparsities of $\{60\%, 70\%, 80\%, 90\%, 95\%\}$ for the decoder cases, and $\{20\%, 40\%, 60\%, 70\%, 80\%, 90\%, 95\%\}$ for the encoder cases. [1]

Before delving into the detailed answers, we briefly address the first question: the impact of depth pruning when applied to the encoder and decoder, respectively. As depicted in Figure 2, depth pruning exhibits a significant influence on the decoder (as indicated by △ and ○), while the encoder shows negligible effects (as observed in △ and ○). Consequently, the appropriate utilization of depth pruning becomes crucial. In the following paragraphs, we outline our key findings related to the second question to establish an effective structured pruning mechanism for encoder-decoder LMs.

**Finding 3.1.** *The number of layers in the decoder network is the dominant factor affecting the inference speed, while the decoder width does not have a significant impact.*

We evaluate the findings regarding the decoder network from two perspectives: (1) the effectiveness of layer-wise pruning and (2) the ineffectiveness of width pruning. Firstly, as demonstrated in the second and fourth plots of Figure 2, the decoder exhibits a significant speedup with minor degradation (when comparing △ and ○), whereas the encoder shows no such effect (when comparing △ and ○). This indicates that layer-wise pruning plays a dominant role in pruning the decoder. On the other hand, when comparing the model size

---

[1] In the case of a high level of target sparsity, we observed that CoFi-T5 is unable to achieve the desired sparsity. A detailed explanation of this phenomenon is in the Appendix A.

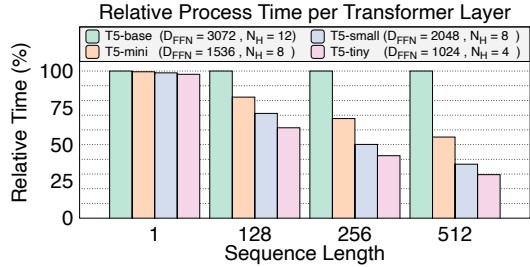

Figure 3: Processing time per one Transformer layer depending on the model configuration and the sequence length. As depicted in the sequence length 1 case, the factors, such as the number of attention heads and FFN dimensions not affect the processing time.

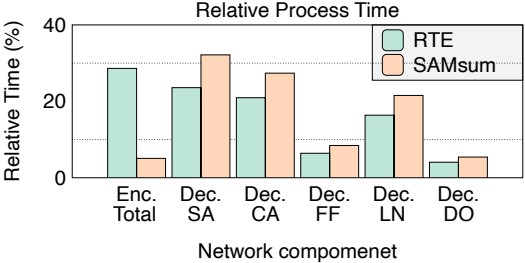

Figure 4: Componentwise processing time of the T5-base. The layer normalization and dropout contribute 20-25% of the total inference time.

and speedup (as shown in the first and second plots of Figure 2 with ⚪), width pruning reduces the model size but leads to both performance degradation and negligible speedup. This suggests that width pruning is not effective for the decoder.

To further investigate Finding 3.1, we investigate the inference speed of Transformer layers, with a specific focus on understanding why width pruning is ineffective. This analysis involves two key observations: (1) finding the metric that synergizes with width pruning, and (2) identifying the component that predominantly consumes computational resources. According to Figure 3, width pruning can have a significant impact on the computational cost as the sequence length increases. However, due to the inherent nature of the autoregressive decoding process, the decoder network is constrained to a sequence length of 1. As a result, width pruning cannot effectively improve the speed of the decoder network. Furthermore, as illustrated in Figure 4, Layer Normalization (LN) and dropout (DO) collectively contribute approximately 20-25% to the overall inference time. Consequently, the time allocated to these fixed operations remains constant, leading to diminished efficiency in terms of inference speed. In conclusion, width pruning is not an appropriate approach for optimizing the decoder.

**Finding 3.2.** *From the perspective of encoder pruning, while achieving high-level sparsity may not be desirable, attaining low-level sparsity not only slightly accelerates inference speed but also enhances performance.*

By comparing the ⚪ points and ★ in the second and fourth plots of Figure 2, we observe that encoder pruning yields a slight speedup along with improved performance. However, when the encoder network is heavily pruned, it experiences significant performance degradation. These findings emphasize the significance of considering pruning in both the decoder and encoder networks. Furthermore, they provide insights into the necessity of employing distinct pruning strategies for these two networks, considering their unique characteristics.

**Comparison with Prior Observations.** Our key findings provide valuable insights: *the appropriate strategy for encoder-decoder models involves using a small number of layers for the decoder and minimal pruning for the encoder networks.* Importantly, our observations offer a more generalized understanding compared to previous works (Kasai et al., 2020; Tay et al., 2021). Unlike prior studies that manually determined model configurations for specific tasks such as machine translation (Kasai et al., 2020) or NLU (Tay et al., 2021), our conclusions are derived automatically through gradual structured pruning and have been validated across both NLG and NLU tasks. Furthermore, while the DeepNarrow strategy proposed by Tay et al. (2021) demonstrates effectiveness in NLU tasks with short output sequences, it exhibits computational inefficiency when applied to NLG tasks. Similarly, the contribution of processing time for encoder networks varies, necessitating the use of a narrower encoder architecture contrary to the approach proposed by Kasai et al. (2020).

## 4  Narrow Encoder and Shallow Decoder

Based on the findings presented in Section 3, we propose a structured pruning framework called **NASH** (**Na**rrow encoder and **Sh**allow decoder) that is specifically optimized for encoder-decoder LMs. Our approach focuses on enhancing inference speed by utilizing uniform layer selection in the decoder network, deviating from the gradual pruning technique commonly employed in encoder-only models. Additionally, we improve generation per-

formance by applying gradual $L0$ regularization pruning specifically to the encoder network, inducing low sparsity instead of solely prioritizing inference speed improvement.

### 4.1 Shallow Decoder: Uniform Layer Selection for Decoder Networks

For a given number of selected layers $d_s$, we can generate a sub-network of the decoder network with a set of selected layers as follows:

$$L_s = \left\{ \left\lfloor \frac{L-1}{d_s-1} \right\rfloor \cdot \ell + 1 \middle| \ell \in \{0, \ldots, d_s - 1\} \right\}$$

We match the hidden states of the sub-networks to those of unpruned decoder networks:

$$\mathcal{L}_{\text{h}}^{\text{dec}} = \sum_{\ell \in \{1,\ldots,d_s\}} \text{MSE}(\mathbf{H}_{\text{dec},s}^{\ell}, \mathbf{H}_{\text{dec},t}^{\left\lfloor \frac{L-1}{d_s-1} \right\rfloor \cdot \ell + 1}).$$

While uniformly selecting layers work well on various domains such as knowledge distillation (Jiao et al., 2019; Shleifer and Rush, 2020) or structured pruning of encoder-only model (Hou et al., 2020), our work first proposes using uniform layer selection of decoder network for structured pruning of encoder-decoder LMs.

The key philosophy of our proposed module is twofold: (1) As shown in Finding 3.1, the number of layers in the decoder network is the main factor affecting inference speedup. (2) Uniform selection is proven to be an effective approach for selecting layers (Hou et al., 2020). To verify this second statement, we compare various candidates, including uniform selection, selection with lower layers, selection with higher layers, and the $L0$ regularization-based approach (Louizos et al., 2018). Through our empirical evaluation, we confirm that uniform selection is the best approach among these candidates (see Section 5.3 and Table 3 for details). Based on this philosophy, we construct shallow decoder pruning by selecting the layers using uniform selection.

### 4.2 Narrow Encoder: Gradual $L0$-regularization with Low Sparsity

Among various structured pruning methods (Hou et al., 2020; Lagunas et al., 2021), we utilize the $L0$ regularization-based pruning method, which has shown the state-of-the-art performances in encoder-only language models (Wang et al., 2020c; Xia et al., 2022). The application of $L0$ regularization

in practice is achieved by enforcing an equality constraint between the target sparsity and the current sparsity:

$$\mathcal{R} = \lambda_1(\hat{s} - t) + \lambda_2(\hat{s} - t)^2,$$

where $\lambda_1$, $\lambda_2$, $\hat{s}$, and $t$ denote the learnable Lagrange multipliers, current sparsity, and target sparsity, respectively. The detailed derivation of $\mathcal{R}$ is described in Appendix B. The current sparsity, $\hat{s}$, is calculated as follows:

$$\hat{s} = \frac{4}{M} \cdot d_h \cdot \sum_i^L \sum_j^{N_h} \mathbf{z}_{\text{head}}^{(i,j)} + \frac{2}{M} \cdot \sum_i^L \sum_j^{d_f} \mathbf{z}_{\text{int}}^{(i,j)},$$

where $M$, $L$, $N_h$, and $d_f$ indicate the number of model parameters, encoder layers, attention heads, and feed-forward layer dimensions, respectively.

We only conduct the pruning of individual attention heads and intermediate layer dimensions by introducing variables $\mathbf{z}_{\text{head}}^{(i)}$ and $\mathbf{z}_{\text{int}}$.

$$\text{MHA}(\mathbf{Q}, \mathbf{K}, \mathbf{V}) = \sum_{i=1}^{N_h} \mathbf{z}_{\text{head}}^{(i)} \text{MHA}_i(\mathbf{Q}, \mathbf{K}, \mathbf{V}) \mathbf{W}_O^{(i)},$$

$$\text{FFN}(\mathbf{X}) = \text{GELU}(\mathbf{X}\mathbf{W}_1) \cdot \text{diag}(\mathbf{z}_{\text{int}}) \cdot \mathbf{W}_2.$$

We further use hidden states distillation by matching the hidden states of pruned and unpruned networks at the same layers as follows:

$$\mathcal{L}_{\text{h}}^{\text{enc}} = \sum_{\ell \in \{1,\ldots,L\}} \text{MSE}(\mathbf{H}_{\text{enc},s}^{\ell}, \mathbf{H}_{\text{enc},t}^{\ell}).$$

As we demonstrated in Finding 3.2, structured pruning with low sparsity enables output quality enhancement rather than inference speedup gain. Motivated by this finding, unlike previous methods (Wang et al., 2020c; Xia et al., 2022) that mainly use $L0$ regularization to achieve high inference speedup, we use such $L0$ regularization to accomplish improvement of output quality.

### 4.3 Training Loss Function of NASH

We combine hidden states distillation with prediction-layer distillation by using Kullback–Leibler divergence (KLD) function.

$$\mathcal{L}_{\text{pred}} = \text{KLD}\left(f(\cdot) \| g(\cdot)\right),$$

where the $f(\cdot)$ and $(\cdot)$ are softmax outputs for the sub-network of pruned model and unpruned model,

Table 1: The summary of Figure 5 which compares the generation quality and latency speedup of **NASH** against other acceleration methods on TweetQA (Xiong et al., 2019), XSum (Narayan et al., 2018), SAMSum (Gliwa et al., 2019), and CNN/DailyMail (See et al., 2017). The numbers of parameters for all models are around 60M, except for T5-Base. The best and second-best results of sharing the dataset are highlighted in bold and underline.

| Task | Question Answering | | Summarization | | | | | |
|---|---|---|---|---|---|---|---|---|
| Dataset | TweetQA | | XSum | | SAMSum | | CNN/DailyMail | |
| | METEOR | Speedup | ROUGE-L | Speedup | ROUGE-L | Speedup | ROUGE-L | Speedup |
| T5-Base | 57.16 | 1.00× | 28.66 | 1.00× | 43.33 | 1.00× | 37.47 | 1.00× |
| T5-Small | 51.42 | 1.92× | 25.04 | 1.96× | 38.14 | 2.14× | 35.59 | 2.11× |
| CoFi-T5* | 49.99 | 1.42× | 25.04 | 1.12× | 37.35 | 1.45× | 34.71 | 1.53× |
| **NASH** (2 decoder layers) | 48.03 | **4.02×** | 26.96 | **3.74×** | 38.89 | **5.53×** | 33.70 | **4.91×** |
| **NASH** (3 decoder layers) | 50.77 | 2.99× | 28.20 | 2.42× | 41.09 | 4.24× | 36.02 | 3.37× |
| **NASH** (4 decoder layers) | **55.08** | 2.52× | **28.64** | 1.64× | **41.34** | 2.99× | **36.78** | 2.69× |

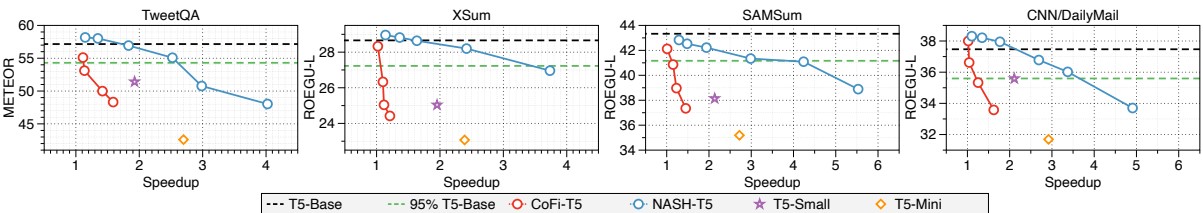

Figure 5: METEOR (or ROUGE-L) vs. speedup on abstractive question answering (TweetQA) and summarization (XSum, SAMSum, and CNN/DailyMail) tasks. We compare **NASH** on 220M T5-Base against CoFi-T5 on T5-Base, 60M T5-Small, 31M T5-Mini (Tay et al., 2021). On all datasets, **NASH** is able to outperform CoFi-T5.

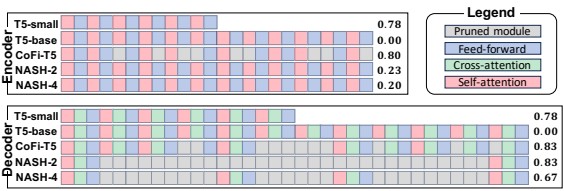

Figure 6: Comparison of the unpruned layers between **NASH** and the other methods on SAMSum. The values on the right of each row indicate the corresponding sparsity compared to the part (encoder or decoder) of T5-base, respectively.

respectively. Then, the total training objective for a pruned model is

$$\mathcal{L} = \mathcal{L}_{\text{pred}} + \lambda^{\text{enc}} \mathcal{L}_{\text{h}}^{\text{enc}} + \lambda^{\text{dec}} \mathcal{L}_{\text{h}}^{\text{dec}} + \mathcal{R},$$

where $\lambda^{\text{enc}}$ and $\lambda^{\text{dec}}$ are coefficients for controlling the contribution of hidden state distillation for the encoder and decoder network, respectively.

## 5 Experiments

### 5.1 Experimental Setup

**Dataset.** We evaluate our proposed method on various tasks using the versatility of *encoder-decoder* LMs. For abstractive question answering, we conduct experiments on TweetQA (Xiong et al.,

2019). For the text summarization task, we experiment on XSum (Narayan et al., 2018), SAMSum (Gliwa et al., 2019), and CNN/DailyMail (See et al., 2017). We evaluate the output quality using METEOR (Banerjee and Lavie, 2005) for abstractive question answering and ROUGE (Lin, 2004) for the summarization tasks. We conduct experiments on multi-task scenario that consists of SAMSum, TweetQA, and five tasks from GLUE (Wang et al., 2018) and SuperGLUE (Wang et al., 2019) benchmarks. The detailed explanations for the datasets used are described in Appendix C.

**Implementation.** First, we fine-tune a model and perform uniform layer selection on the decoder network of the fine-tuned model to generate a sub-network model. Subsequently, we further train the sub-network model with the pruning objective, utilizing a scheduler to gradually increase the sparsity until reaching the desired target value. In our experiments, we calculate sparsity by dividing the number of pruned parameters (excluding the embedding) by the size of the full model. Following the approach of Xia et al. (2022), we continue fine-tuning the pruned model until convergence. We set the target sparsity of the encoder networks as 30% for all experiments. The reported results are based on

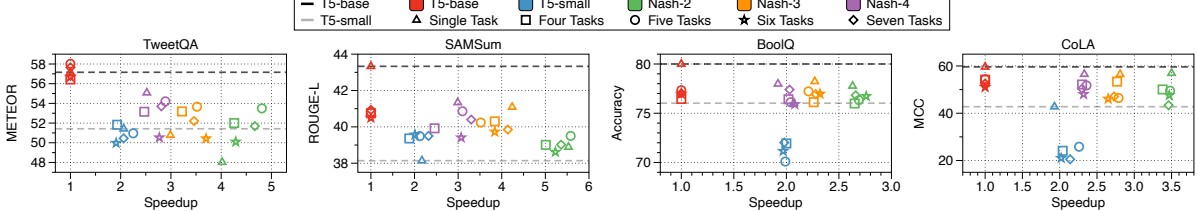

Figure 7: Evaluation of generation quality and latency speedup of **NASH** on multi-task learning scenario. We compare **NASH** on 220M T5-Base against T5-Base and 60M T5-Small. On all datasets, **NASH** is able to outperform T5-Small.

Table 2: Evaluation of generation quality and latency speedup of **NASH** on instruction-tuning scenario. Note that **NASH**-$L$ indicates that we conduct our **NASH** with decoder layer number of $L$.

| Method | Dolly Evaluation | | | Self-Instruct | | | Vicuna Evaluation | | |
|---|---|---|---|---|---|---|---|---|---|
| | GPT Eval | ROUGE-L | Speedup | GPT Eval | ROUGE-L | Speedup | GPT Eval | ROUGE-L | Speedup |
| T5-Base | 61.32 | 32.53 | 1.00× | 40.15 | 14.20 | 1.00× | 59.26 | 20.47 | 1.00× |
| T5-Small | 47.03 | 29.29 | 1.96× | 29.82 | 11.67 | 1.91× | 34.98 | 17.40 | 2.21× |
| **NASH**-2 | 41.28 | 28.09 | 5.15× | 26.74 | 10.58 | 5.27× | 45.53 | 20.09 | 5.90× |
| **NASH**-3 | 52.82 | 30.87 | 3.62× | 32.18 | 11.91 | 3.79× | 50.59 | 21.07 | 4.24× |
| **NASH**-4 | 56.85 | 31.43 | 2.67× | 36.44 | 13.01 | 2.94× | 54.21 | 21.83 | 3.17× |

the validation sets of all datasets. Additionally, our models are implemented using Huggingface (Wolf et al., 2020) library.

## 5.2 Main Results

**Standard Fine-tuning.** Table 1 and Figure 5 summarize that the proposed method outperforms CoFi-T5, 6-layer T5-Small,(Raffel et al., 2020), and 4-layer T5-mini,(Tay et al., 2021) in terms of output quality and inference speedup. Our results consistently demonstrate the superiority of **NASH** with three decoder layers over the baselines in both performance metrics, although there is a trade-off between output quality and inference speedup with our method. Moreover, our method is particularly effective in improving inference speed, especially for tasks involving longer sequence outputs, such as SAMSum or CNN/DailyMail. However, CoFi-T5 falls short in achieving comparable speedups to T5-Small while maintaining the same output quality. Figure 6 illustrates the pruned model structure trained on SAMSum, following the method described in Table 1 to provide a deeper understanding of the results. Despite CoFi-T5 removing more modules in the encoder network compared to **NASH**, it cannot remove as many layers in the decoder layer as **NASH** does.

**Multi-task Learning.** We conducted an analysis to observe how performance trend change with varying numbers of tasks. As depicted in Figure 7, both fine-tuned T5-Base and our algo-

rithm (**NASH**) exhibit fluctuations in accuracy according to the number of tasks. However, it is worth noting that our proposed method demonstrates significantly less variation in generation performance when compared to T5-Small. This robustness in multi-task learning is consistently observed across all datasets. Overall, **NASH** consistently outperforms T5-Small in terms of both generation performance and speedup, similar to standard fine-tuning. Through the results shown in complex scenarios, we verify that the generation performance of encoder-decoder models is robust to the number of decoder layers.

**Instruction Tuning.** To verify versatility of proposed **NASH**, we consider the instruction-tuning scenario with the databricks-dolly-15k (Conover et al., 2023) dataset. By following Gu et al. (2023), we split the total dataset into 14k train samples and 1k evaluation samples. We evaluate the trained model (with 14k of train samples) on three instruction-following datasets to check the generalizability of the model across the different datasets: (1) 1k dolly evaluation; (2) Self-Instruct (Wang et al., 2023a); (3) Vicuna evaluation (Chiang et al., 2023). Similar to previous instances of task-specific fine-tuning and multi-task scenarios, our algorithm with three decoder layers consistently outperforms T5-Small across various metrics, including GPT-4 evaluation, ROUGE-L, and speedup, as shown in Table 2. These results suggest that our proposed method is well-suited for developing general-

Table 3: The performance of various layer selection strategies on different datasets is presented. "Gray" indicates a failure to achieve the target sparsity. Additionally, we report the number of remaining SA, CA and FF layers for the automatic selection method.

| Task | RTE | BoolQ | CB | SAMSum | TweetQA |
|---|---|---|---|---|---|
| Uniform (Ours) | **76.17** | 79.51 | **91.07** | **47.37** | **52.74** |
| Low | 75.81 | **79.82** | 89.29 | 46.14 | 49.70 |
| High | 75.45 | 78.99 | 87.50 | 39.45 | 47.40 |
| Louizos et al. (2018) | 78.70 | 79.20 | 92.86 | 48.52 | 56.41 |
| # SA, CA, FF | 2,4,2 | 0,3,2 | 4,7,4 | 6,7,5 | 11,10,8 |

Table 4: Comparison of pruning strategy on encoder network. We conduct our method on T5-base with a uniform selection of 4 decoder layers.

| Dataset | RTE | | SAMSum | | TweetQA | |
|---|---|---|---|---|---|---|
| | Acc | Speedup | R-L | Speedup | MTR | Speedup |
| None | 77.97 | 2.12× | 41.06 | 2.78× | 54.73 | 2.42 × |
| Unif. (S 0.25) | 74.36 | 2.35× | 40.42 | 2.86× | 53.62 | 2.55 × |
| Unif. (S 0.5) | 71.84 | 2.59× | 39.36 | 2.94× | 49.62 | 2.68 × |
| $L0$ Reg. (S 0.3) | **78.70** | 2.24× | **41.34** | 2.99× | **55.08** | 2.52× |
| $L0$ Reg. (S 0.6) | 76.89 | **2.61×** | 40.49 | **3.12×** | 49.48 | **2.75×** |

Table 5: Comparison of results achieved by CoFi-T5 and NASH on deeper models (Tay et al., 2021) and SAMSum.

| Task | NL 16 | | NL 20 | | NL 24 | |
|---|---|---|---|---|---|---|
| | R-L | Speedup | R-L | Speedup | R-L | Speedup |
| Base | 41.31 | 1.0× | 41.73 | 1.0× | 41.83 | 1.0× |
| CoFi-T5* | 33.53 | 1.6× | 35.87 | 1.6× | 36.65 | 1.5× |
| **NASH** (2 DL) | 36.50 | 4.6× | 35.45 | 5.2× | 35.29 | 5.9× |
| **NASH** (4 DL) | 39.05 | 3.6× | 38.66 | 3.8× | 38.25 | 4.3× |

Table 6: Comparison of results of **NASH** and Tao et al. (2023) on BART-Base and CNN/DailyMail. Results of Tao et al. (2023) are from the original work.

| | BART Base | **NASH** | | Tao et al. (2023) | |
|---|---|---|---|---|---|
| | | **NASH**-2 | **NASH**-3 | 50% | 27% |
| ROUGE-L | 41.68 | 40.87 | **41.14** | 39.91 | 40.39 |
| Speedup | 1.0× | **3.2×** | 2.1× | ~1.5× | |
| # Params | 139M | 80M | 89.8M | **70.9M** | 102.4M |

purpose language models, a usage pattern widely adopted in recent large language models (Chung et al., 2022a; Tay et al., 2023).

## 5.3 Ablation Studies

**Different Layer Selection.** To validate the effectiveness of uniform layer selection in shrinking the decoder network, we investigate other layer selection methods in a two-layer selection problem. We compare four selection methods: lower selection (first 2 layers), higher selection (last 2 layers), L0 regularization-based automatic selection (Louizos et al., 2018; Xia et al., 2022), and our uniform selection. The results presented in Table 3 demonstrate that uniform selection consistently outperforms the other manual selection methods. The performance margin becomes more pronounced in NLG tasks. Notably, we observe that automatic selection fails to achieve the target sparsity consistently across all tasks, except for BoolQ. This instability in automatic selection aligns with our preliminary findings discussed in Appendix A.

**Different Pruning on Encoder.** To evaluate the effectiveness of our pruning strategy on the encoder network, we investigate the following strategies at different levels of sparsity: (1) without encoder pruning, (2) uniform layer selection (similar to the decoder network), and (3) the proposed $L0$ regularization approach. We prune the encoder network of the T5-Base, which has four decoder layers selected uniformly. The results presented in Table 4 clearly demonstrate that our chosen approach, with low sparsity, outperforms both the unpruned baseline and the uniform layer selection. We also observe that the advantage of this approach is only noticeable at low sparsity, as evidenced by the comparison between 30% and 60% sparsity.

**NASH on Different LMs.** We also conducted a deeper model experiment using T5-Small-efficient,(Tay et al., 2021), which is a variant of T5-Small with up to four times more layers while maintaining the same configuration. This experiment aimed to determine the effectiveness of our method regardless of the model architecture. The results presented in Table 5 consistently demonstrate that **NASH** improves inference speed without significantly compromising the quality of generated outputs, regardless of the depth of the decoder networks. It is noteworthy that the acceleration compared to the original model increases as the number of decoder layers increases. Furthermore, **NASH** exhibits faster inference and higher output quality compared to CoFi-T5, which is consistent with the results presented in Table 1.

**Comparison with Tao et al. (2023).** We applied **NASH** to BART-Base (Lewis et al., 2020) using the CNN/DailyMail dataset, conducting a direct comparison with SIMPLE (Tao et al., 2023). SIMPLE introduced a structured pruning method for generative LMs, which is relevant to our work. Notably, **NASH** exhibits higher ROUGE-L scores than SIM-PLE when both models are at 27% sparsity. Additionally, despite having larger parameters, **NASH**

outperforms SIMPLE with 50% sparsity in terms of speedup. Our approach achieves more than three times the speedup, while SIMPLE reaches a maximum of 1.5 times on the GPU.

# 6 Related Works

**Language Model Compression.** With the advancement of NLP, LMs have grown in size, making it difficult to deploy them on edge devices and resulting in slower inference speed. As a result, there has been active research on language model compression which has three main approaches: quantization, knowledge distillation, pruning. Quantization (He et al., 2016; Alom et al., 2018; Zafrir et al., 2019; Shen et al., 2020; Yao et al., 2022) minimizes the storage requirements for weight values by reducing the number of bits needed to represent them. Knowledge distillation (Sanh et al., 2019; Jiao et al., 2019; Sun et al., 2019, 2020; Wang et al., 2020b,a) transfers the knowledge of a large-scale teacher model with high performance to a smaller-scale student model, enabling the student model to replicate the behavior of the teacher model. Pruning (Chen et al., 2020; Sanh et al., 2020; Kwon et al., 2022; Frantar and Alistarh, 2023) reduces the size of a model by removing unnecessary parts of large networks such as neurons, weights, or layers.

**Pruning.** Pruning can be categorized into two parts: (1) unstructured pruning and (2) structured pruning. In unstructured pruning (Chen et al., 2020; Prasanna et al., 2020), weights, which are connections between neurons, are removed from the network based on various criteria. However, this line of methods produces sparse weight matrices, requiring specific hardware support. On the other hand, structured pruning (Xia et al., 2022; Kwon et al., 2022; Kurtic et al., 2023), prunes away structures such as neurons, weight matrix blocks, or layers. Most previous works on structured pruning have focused on encoder-based models (Xia et al., 2022; Kwon et al., 2022; Kurtic et al., 2023), which remove attention heads, columns, and rows of weight matrices using different importance score metrics, including magnitudes or Hessians of weight matrices, and $L0$ loss. However, structured pruning on generative models has been significantly under-investigated, with only a few available works (Lagunas et al., 2021; Yang et al., 2022; Santacroce et al., 2023). Lagunas et al. (2021) extended movement pruning (Sanh et al., 2020) into structured prun-

ing, but their method can only achieve up to $1.4\times$ speedup for *encoder-decoder* based BART (Lewis et al., 2020). Yang et al. (2022) released an open-source toolkit that combines structured pruning and vocabulary pruning for various pre-trained language models, but only vocabulary pruning is applicable to T5 and BART.

# 7 Conclusion

We propose **NASH** to address the lack of exploration in structured pruning of *encoder-decoder* LMs. To design a structured pruning method suitable for encoder-decoder models, we first examine the behavior of pruned models with different strategies, focusing on inference speed and generation performance. Our findings reveal that (1) the number of decoder network layers is the key factor in accelerating inference speed and (2) low sparsity pruning on the encoder network can enhance model performance. Based on these insights, we develop **NASH**, which constructs a narrow encoder and a shallow decoder network for *encoder-decoder* LMs through gradual $L0$ regularization pruning and uniform layer selection, respectively. We demonstrate the superiority of **NASH** in terms of speedup and output quality across various tasks. We strongly believe this work lays a strong foundation for further investigation into effective pruning approaches for encoder-decoder LM.

# Limitations

Although we were unable to conduct research on unstructured pruning due to device limitations, collaboration with devices could facilitate performance enhancements. Furthermore, owing to the motivating analysis and algorithm construction of this paper, *i.e.,* analysis of separate encoder and decoder networks, further exploration of a co-optimized method is necessary, and there is potential for improvement in this aspect.

# Acknowledgment

This work was supported by the "Research on model compression algorithm for Large-scale Language Models" project funded by KT (KT award B220002586, 90%) and Institute of Information & communications Technology Planning & Evaluation (IITP) grant funded by Korea government (MSIT) [No. 2019-0-00075, Artificial Intelligence Graduate School Program (KAIST), 10%].

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

Learning Representations*.

## A  Instability of CoFi-T5

In this section, we observe that pruning of the T5 shows more varied accuracy and pruned sparsity than those of BERT under the same condition of pruning (e.g., target sparsity, warm-up epochs) which means instability of encoder-decoder model pruning.

**Experimental Setup.**  We study the instability of CoFi-T5 with T5-Base compared to the original CoFi with BERT-Base (Devlin et al., 2019). To compare the T5 and BERT, we conduct the experiments on the RTE task of the GLUE benchmark (Wang et al., 2018) with 90% of target sparsity[2] for both models. Additionally, we extend our investigation to the SAMSum (Gliwa et al., 2019) which contains messenger-like conversations with summaries, utilizing T5-Base to observe the training instability of the encoder-decoder model on more challenging tasks. We prune with 20 random seeds to compare different settings.

**Results.**  Figure 8 presents two main observations. *Firstly*, the output performance of pruned T5 models exhibits more variability than pruned BERT models at high-level target sparsity. *Secondly*, CoFi-T5 fails to achieve the target sparsity at high-level sparsity in both RTE and SAMSum. In the case of RTE, we observed a high proportion of over-pruning, while in the case of SAMSum, all experiments were significantly under-pruned. These results indicate the instability of CoFi-T5 in terms of sparsity and output quality, which aligns with the findings discussed by Zhang et al. (2021).

$L0$  regularization-based  structured  pruning methods (Wang et al., 2020c; Xia et al., 2022) commonly incorporate linear warm-up to gradually increase the target sparsity during the training process, aiming to ensure stable training of mask variables. Based on this understanding, we employ longer warm-up epochs for gradual structured pruning and observe that this approach partially mitigates the instability of CoFi-T5. While this mitigates the training instability to some extent, it does not completely address the challenge associated with CoFi-T5. These results motivate us to investigate the appropriate strategy for structured pruning in encoder-decoder models. [3]

---

[2]The sparsity is computed as the number of pruned parameters divided by the full model size (embeddings and classifier excluded).

[3]As longer warm-up epoch is shown to be effective, we applied it to both CoFi-T5 and NASH in our experiments.

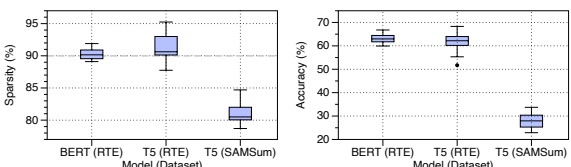

Figure 8: The pruned sparsity (left) and output performance (right) distributions for 90% of target sparsity across 20 random trials on RTE (Wang et al., 2018) with BERT-Base, RTE with T5-Base, and SAMSum (Gliwa et al., 2019) with T5-Base. Unlike the results of BERT on RTE, which show that the accuracy and pruned sparsity are concentrated around the target sparsity, we observe that both output performance and pruned sparsity vary across different trials. This variation confirms the instability of pruning-aware training for encoder-decoder models. It is important to note that accuracy and ROUGE-L are used as output performance metrics for RTE and SAMSum, respectively.

## B  Pruning with $L0$ Regularization

In this section, we give the details of the pruning with $L0$ regularization. Structured pruning through $L0$ regularization (Louizos et al., 2018) is proposed to construct the sparse neural network efficiently. In addition, this scheme using $L0$ regularization is widely applied to prune LMs (Xia et al., 2022; Wang et al., 2020c). With a given LM $f(\cdot; \theta)$ that is parameterized by $\theta = \{\theta_j\}_{j=1}^n$, we can define binary mask variable $\mathbf{z} = \{z_j\}_{j=1}^n$. Note that $\theta_j$ and $z_j$ denote a unit of weights (e.g., weights of attention heads, or column of an MLP layer) and mask variable corresponding to $\theta_j$, respectively.

In this formulation, a pruned LM is written as $f(\cdot; \tilde{\theta})$, where $\tilde{\theta} = \theta \odot \mathbf{z}$ and the pruning is a problem to find optimal mask variables and weights. In the $L0$ regularization-based pruning, these masks and weights are learned based on the following objective function:

$$\min \mathbb{E}_z \left[ \frac{1}{D} \sum_{i=1}^{D} \mathcal{L}(f(x_i; \tilde{\theta}), y_i) + \lambda ||\tilde{\theta}||_0 \right]$$

In the objective function above, every mask variable $z_j$ is chosen based on the prior distribution $q(z_j|\pi_j) = \text{Bernoulli}(\pi_j)$. Considering the number of binary masks, n, the possible choices of z can be represented as $2^n$. The discrete feature of the mask and this tremendous amount of choices make mask optimization practically intractable.

Louizos et al. (2018) mitigate this issue with a re-parameterization trick, enabling z to be differentiable and updates with the model parameter $\theta$. In detail, the masks $\mathbf{z}$ are defined as continuous

Table 7: Description and label of NLU datasets in GLUE and SuperGLUE benchmarks.

| | Task | Description | Label |
|---|---|---|---|
| GLUE | CoLA | To determine the linguistic acceptability of a single sentence | ['acceptable', 'not_acceptable'] |
| | MRPC | To answer whether human annotators paraphrase input sentence pairs | ['equivalent', 'not_equivalent'] |
| | STS-B | To answer how much input sentence pairs are semantically similar | 0<x<5 (regression) |
| | RTE | To determine the logical relationship between input sentence pairs | ['entailment', 'not_entailment'] |
| | SST-2 | To answer whether the input sentence contains positive or negative sentiment | ['negative','positive'] |
| | QNLI | To determine whether a given context entails the answer to a corresponding question | ['entailment', 'not_entailment'] |
| SGLUE | CB | To determine the linguistic acceptability of a single sentence | ['acceptable', 'not_acceptable'] |
| | COPA | To comprehend the logical connections between events and make accurate judgments | ['choice1', 'choice2'] |
| | WiC | To determine the linguistic acceptability of a single sentence | ['true', 'false'] |
| | BoolQ | to determine the linguistic acceptability of a single sentence | ['true', 'false'] |

variables determined by $\min(1, \max(0, \mathbf{s}))$, where continuous random variable $\mathbf{s}$ is sampled from the range of $[0, 1]$. Note that it is equivalent to sample u from the uniform distribution, $U(0, 1)$ and calculate s as follows:

$$\mathbf{s} = \text{sigmoid}(\log\mathbf{u} - \log(1 - \mathbf{u}) + \alpha),$$
$$\bar{\mathbf{s}} = \mathbf{s} \times (r - l) + l, \mathbf{z} = \min(1, \max(0, \bar{\mathbf{s}}))$$

where $l$ and $r$ are constant values that satisfy $l < 0$ and $r > 0$, and $\alpha$ is a learnable parameter. From this formulation, the learning objective can be rewritten as:

$$\min \mathbb{E}_{u \sim U(0,1)} \left[ \frac{1}{D} \sum_{i=1}^{D} \mathcal{L}(\mathbf{p}(x_i; \tilde{\theta}), y_i) + \lambda||\tilde{\theta}||_0 \right]$$

This process obtains $\mathbf{z} = \{z_j\}_{j=1}^n$ where every $z_j$ is in the range of [0, 1]. However, Wang et al. (2020c) observes that optimizing with those relaxed regularizations makes models converge to different-size subnetworks depending on a learning rate and pruning schedule. To mitigate this problem, they suggest using a Lagrangian relaxation instead of the $L0$ regularizer $\lambda||\tilde{\theta}||_0$ as follows:

$$\mathcal{R} = \lambda_1(\hat{\mathbf{s}} - t) + \lambda_2(\hat{\mathbf{s}} - t)^2$$

where $\lambda_1$ and $\lambda_2$ are learnable Lagrange multipliers. $\hat{\mathbf{s}}$ represents the current sparsity calculated by the masks $\mathbf{z}$, while $t$ represents the target sparsity. Motivated by these works, we also utilize the relaxed regularization term, $\mathcal{R}$, for gradually structured pruning on the encoder network.

## C Description of Datasets

**Natural Language Generation Tasks.** Since we study the structured pruning for encoder-decoder models, a sort of generative model, we conduct comprehensive experiments on the NLG tasks. The NLG datasets used in our study encompass

two tasks: summarization and abstract question answering. We employed the XSum (Narayan et al., 2018), SAMSum (Gliwa et al., 2019), and CNN/DailyMail (See et al., 2017) datasets to assess the summarization capability of our proposed method. These datasets are widely used in evaluating the effectiveness of summarization techniques. Regarding abstractive question answering, we employed the TweetQA (Xiong et al., 2019) dataset to evaluate our method.

- **XSUM** (Summarization): XSUM (Narayan et al., 2018) comprises articles sourced from BBC, along with corresponding single sentence summaries. More specifically, each article begins with an introductory sentence that serves as a summary. These summaries are professionally written and are usually provided by the article's author.

- **SAMSum** (Summarization): SAMSum (Gliwa et al., 2019) consists of 16K messenger-like conversations annotated with a summary for providing a concise overview of the conversation's content in third person. The conversations encompass a variety of styles and registers, ranging from informal to semi-formal and formal. Additionally, they may include slang words, emoticons, and typographical errors.

- **CNN/DailyMail** (Summarization): CNN / DailyMail (See et al., 2017) consists of over 300K English news articles that were originally designed for machine-reading and comprehension as well as abstractive question answering, but it now also supports extractive and abstractive summarization. In this work, we utilize the 3.0.0 version.

- **TweetQA** (Abstract Question Answering): TweetQA (Xiong et al., 2019) is the first large-scale dataset for question answering (QA) over

Table 8: Hyperparameters for NLG datasets

| Hyperparameter | Value & Description |
|---|---|
| Training epochs | 20 (TQA, SAMSum), 3 (XSum, CNN/DM) 
 how many epochs are trained. |
| Learning rate | $3 \times 10^{-5}$ 
 learning rate by AdamW optimizer |
| Evaluation steps | 1000 
 evaluation frequency |
| Batch size | 4 
 quantity of samples per update |
| Max input length | 512 
 the maximum length for the training |
| Max target length | 128 
 maximum length to be generated |
| Warm-up epochs | 16 (TQA,SAM), 2 (XSum, CNN/DM) 
 epochs that target sparsity meets |
| Reg learning rate | 0.01 
 learning rate for the $\lambda_1$ and $\lambda_2$ in $L0$ regularization |
| $\lambda^{enc}, \lambda^{dec}$ | 0.001 
 weight for layer distill loss |
| Fine-tune epochs | 20 (TQA, SAMSum), 3 (XSum, CNN/DM) 
 epochs for fine-tuning after the pruning |

Table 9: Hyperparameters for NLU datasets

| Hyperparameter | Value & Description |
|---|---|
| Training epochs | 150 (small), 100 (middle), 20 (high) 
 how many epochs are trained. |
| Learning rate | $3 \times 10^{-5}$ 
 learning rate by AdamW optimizer |
| Evaluation steps | 20 (small), 50 (middle), 500 (high) 
 evaluation frequency |
| Batch size | 32 
 quantity of samples per update |
| Max input length | 128 
 the maximum length for the training |
| Max target length | 5 
 maximum length to be generated |
| Warm-up epochs | 120 (small), 80 (middle), 16 (high) 
 epochs that target sparsity meets |
| Reg learning rate | 0.01 
 learning rate for the $\lambda_1$ and $\lambda_2$ in $L0$ regularization |
| $\lambda^{enc}, \lambda^{dec}$ | 0.001 
 weight for layer distill loss |
| Fine-tune epochs | 20 
 epochs for fine-tuning after the pruning |

social media by addressing that previous QA datasets have concentrated on formal text like news and Wikipedia.

**Natural Language Understanding Tasks.** We apply GLUE (Wang et al., 2018) and SuperGLUE benchmarks (Wang et al., 2019) for evaluating on NLU tasks. While these benchmarks consist of classification datasets, we generate the phrase related to the label instead of the class index. The detailed descriptions and labels of each task are described in Table 7.

- **GLUE**: GLUE (Wang et al., 2018) is a collection of datasets for evaluating the performance

of models across a diverse set of existing NLU tasks. By including tasks with limited training data, GLUE is designed to favor and encourage models that share general linguistic knowledge across tasks. In this work, we employ six tasks in GLUE benchmarks: CoLA, MRPC, STS-B, RTE, SST-2, and QNLI.

- **SuperGLUE**: SuperGLUE (Wang et al., 2019) is a new benchmark styled after GLUE with a new set of more difficult NLU tasks. It incorporates improved resources to address the fact that performance on GLUE has surpassed the level of non-expert humans, thereby indicating the limited potential for further research progress. In this work, we adopt some of SuperGLUE tasks: CB, BoolQ, WiC, and COPA.

**Instruction-Tuning Tasks.** We apply databricks-dolly-15k (Conover et al., 2023), Self-Instruct (Wang et al., 2023a), and Vicuna (Chiang et al., 2023) for evaluating on instruction-tuning tasks.

- **databricks-dolly-15k**: databricks-dolly-15k (Conover et al., 2023) is an open-source dataset of instruction-following records generated by thousands of Databricks employees in several of the behavioral categories outlined in the InstructGPT (Ouyang et al., 2022), including brainstorming, classification, closed QA, generation, information extraction, open QA, and summarization.

- **Self-Instrcut**: The authors of Self-Instruct (Wang et al., 2023a) have introduced a dataset comprising 52,000 instructions matched with 82,000 instance inputs and outputs. This dataset serves as a resource for fine-tuning language models to improve their adherence to instructions. Additionally, they've provided 252 expert-created tasks and instructions designed for user-centric applications, which are used in the human evaluation section of their research. Furthermore, the Self-Instruct dataset includes 50,000 examples from the P3 and Super Natural Instructions datasets for the purpose of facilitating comparisons with existing public datasets.

- **Vicuna**: Vicuna (Chiang et al., 2023) utilized approximately 70,000 multi-round conversations between users and ChatGPT collected

from the ShareGPT website (Geng et al., 2023), which allows sharing of ChatGPT dialogues, as a dataset for fine-tuning. In this work, we utilize 80 challenging questions used in the Vicuna evaluation.

## D Hyperparameters

In this section, we describe the hyperparameter setup of experiments. We report the hyperparameters that we utilized in Table 8 and 9 for NLG and NLU tasks, respectively. We use different hyperparameter sets for small NLG datasets (TweetQA, SAMSum) and large NLG (XSum, CNN/DailyMail) datasets. Similarly, for NLU tasks, we use different hyperparameters depending on the dataset size. For the small-size NLU datasets (CB, COPA) that the number of samples is smaller than 1,000, we use the hyperparameters (small) described in Table 9. We especially train our model for 150 epochs because the data size is too small to learn the weights for the L0 regularization. We use the hyperparameters (middle) described in Table 9 for the middle-size NLU datasets (MRPC, RTE, STS-B, CoLA, WIC, BOOLQ) that the number of samples is more than $1,000$ but less than $10,000$. We use the hyperparameters (high) described in Table 9 for the large-size NLU datasets (MNLI, QQP, QNLI, SST-2) that the NLU datasets whose size is larger than $10,000$.

## E Instruction-tuning Details

In the evaluation process of GPT-4, feedback is solicited by instructing the model to compare its generated responses with the authentic, reference answers and assign a numerical score ranging from 1 to 10 to each response. Drawing upon the methodology outlined by Gu et al. (2023), we calculate the ratio between the cumulative scores assigned to the model's responses and those of the ground truth answers. Further details regarding the specific prompt employed for this evaluation are presented in Figure 9.

## F Speedup Evaluation Metric

To measure the inference speed, we conducted inference predictions for each dataset and examined configuration using the PyTorch (Paszke et al., 2019) compiled function. This was done on a single server equipped with a NVIDIA GeForce RTX

We would like to request your feedback on the performance of two AI assistants in response to the user instruction and input displayed above.

Please rate the helpfulness, relevance, accuracy, and level of detail of their responses. Each assistant receives an overall score on a scale of 1 to 10, where a higher score indicates better overall performance.

Please first output a single line containing only two values indicating the scores for Assistant 1 and 2, respectively. The two scores are separated by a space.

In the subsequent line, please provide a comprehensive explanation of your evaluation, avoiding any potential bias and ensuring that the order in which the responses were presented does not affect your judgment.

Figure 9: GPT-4 evaluation prompt.

3090 GPU and an AMD EPYC 7502 32-Core Processor CPU. For each inference prediction, we utilized a batch size of 32. Additionally, we generated output sequences using a beam size of 4. The time taken for the measurements included all decoding steps until completion.

Table 10: Comparison with Tay et al. (2021) using same decoder layer number (# DL) on TweetQA (TQA) and SAMSum (SAM).

| # DL | DL 2 | | DL 4 | | DL 6 | | DL 8 | |
|---|---|---|---|---|---|---|---|---|
| Task | TQA | SAM | TQA | SAM | TQA | SAM | TQA | SAM |
| Tay et al. (2021) | 51.16 | 39.30 | 51.67 | 40.96 | 51.68 | 41.07 | 52.67 | 41.18 |
| **NASH** | 48.03 | 38.90 | 55.11 | 41.37 | 56.92 | 42.23 | 58.02 | 42.51 |

## G Additional Experiments

### G.1 Comparison with Efficient-T5

We compare our algorithm, **NASH**, with models designed to have a shallow decoder depth originally from the pre-training stage, as proposed by Tay et al. (2021). In our evaluation, we examine the performance of our method on two tasks, namely TweetQA and SAMSum, using 2, 4, 6, and 8 decoder layers. As shown in Table 10, **NASH** demonstrates superior performance in most cases. This result is noteworthy as our method can construct a small yet effective model without requiring any costs to make the small pre-trained language model.

### G.2 Results on NLU Tasks

We also compare **NASH** to the baseline methods on the GLUE and SuperGLUE benchmarks, which are focused on NLU tasks. Since these tasks involve relatively longer input sequences and shorter

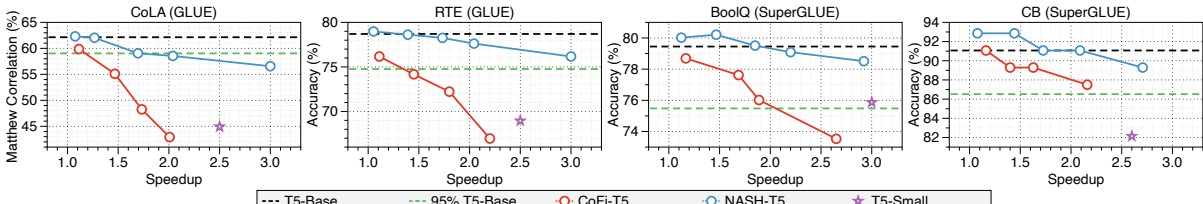

Figure 10: Accuracy vs. speedup on various natural language understanding tasks (CoLA, RTE, BoolQ and CB). We compare **NASH** on 220M T5-Base against CoFi-T5 on T5-Base and 60M T5-Small. On all datasets, **NASH** is able to outperform CoFi-T5.

Table 11: The summary of Figure 10 which compares the understanding accuracy and latency speedup of **NASH** against other acceleration methods on GLUE (Wang et al., 2018) and SuperGLUE (Wang et al., 2019) benchmarks. The number of parameters for all models is around 60M, except for T5-Base. For NASH, we apply two layers of decoder sub-network. The best results of sharing the dataset are highlighted in bold.

| Task | GLUE | | | | | | SuperGLUE | | | |
|---|---|---|---|---|---|---|---|---|---|---|
| Dataset | QNLI | SST-2 | CoLA | MRPC | RTE | STS-B | BoolQ | WiC | COPA | CB |
| T5-Base (Teacher) | 93.26 | 95.53 | 63.14 | 89.71 | 78.70 | 90.85 | 79.45 | 72.57 | 70.00 | 91.07 |
| T5-Small | 91.03 | 92.32 | 44.93 | **88.73** | 68.95 | 89.27 | 75.87 | 69.28 | 55.00 | 82.14 |
| Speedup↗ | 2.5× | 2.8× | 2.5× | 2.4× | 2.3× | 2.6× | **3.0×** | 2.0× | 2.0× | 2.6× |
| CoFi-T5* | 87.88 | 91.28 | 42.92 | 75.74 | 66.96 | 86.49 | 78.69 | 68.97 | 65.00 | 87.50 |
| Speedup↗ | 2.1× | 2.1× | 2.0× | 2.1× | 1.7× | 1.9× | 2.0× | 2.0× | 2.2× | 2.2× |
| **NASH** (Ours) | **91.19** | **92.74** | **56.56** | 86.27 | **76.17** | **90.39** | **79.51** | **71.16** | **69.00** | **89.29** |
| Speedup↗ | **3.0×** | **2.8×** | **3.0×** | **3.3×** | **3.0×** | **2.9×** | 2.9× | **2.6×** | **3.0×** | **2.7×** |

output sequences compared to NLG tasks, our proposed method exhibited less effectiveness. However, **NASH** still demonstrates superiority over the baselines, as depicted in Figure 10. It is important to note that the performance of our method remains robust across different compression rates. We also provide detailed performance results of our proposed method for the full GLUE and SuperGLUE benchmarks in the Appendix G.2 in Table 11. The results demonstrate the effectiveness of our proposed **NASH** method in achieving high output quality while significantly improving inference speed. The superior performance of **NASH** across both GLUE and SuperGLUE benchmarks highlights its potential as an efficient acceleration method for NLU tasks as well as for NLG tasks.