# OpenReview forum: "NASH: A Simple Unified Framework of Structured Pruning for Accelerating Encoder-Decoder Language Models"
_EMNLP/2023/Conference — EMNLP 2023 Findings_

### Official Review · Reviewer_XszA · 2023-08-01

**Soundness:** 3

**Excitement:**

3: Ambivalent: It has merits (e.g., it reports state-of-the-art results, the idea is nice), but there are key weaknesses (e.g., it describes incremental work), and it can significantly benefit from another round of revision. However, I won't object to accepting it if my co-reviewers champion it.

**Paper Topic And Main Contributions:**

the submission proposed a pruning strategy for encoder-decoder models that emphasizes increasing the sparsity of the encoder, and reducing the number of layers in the decoder. The proposal was founded on empirical evidence of the tradeoff among the model size, accuracy, and latency. In addition, improvements are shown in extensive experimental results.

**Reasons To Accept:**

1. sufficient empirical evidence is presented for motivating the proposal.

2. it is the first attempt to propose a systematic strategy to prune encoder-decoder models.

3. extensive ablation study demonstrates the effectiveness of the proposed algorithm.

**Reasons To Reject:**

1. the separation or the improvement of increasing the sparsity of the encoder over solely pruning the decoder doesn't seem significant to me, which then makes the contribution of this submission uncertain, at least to me.  It is generally understood that the number of layers in a decoder-only model plays a large role in the latency, therefore, naturally people would start with pruning the number of layers in the decoder.

2. I wonder if it is possible to distill a pre-trained language model using this proposed strategy to achieve a much simpler but also general-purpose language model rather than a task-specific language model. imo, that'd be much more beneficial to the community since it would essentially reduce the latency for all downstream computation, including finetuning and zero-shot inference.

**Reproducibility:**

4: Could mostly reproduce the results, but there may be some variation because of sample variance or minor variations in their interpretation of the protocol or method.

**Reviewer Confidence:**

4: Quite sure. I tried to check the important points carefully. It's unlikely, though conceivable, that I missed something that should affect my ratings.

---

> ### Author Rebuttal · Authors · 2023-08-29
>
> We would appreciate your insightful comments based on a thorough inspection of our paper. We are encouraged that you found that our paper is well-motivated from sufficient empirical analysis and proposed algorithm is effective. The answers for your concerns are addressed as follows.
>
> ***Q1. The significance and novelty of contributions (width pruning on encoder, depth pruning on decoder)***
>
> **A1.**
> **Summary:**
>
> In this work, we provided several observations that are (1) ***counter-intuitive perspective on the recent direction of structured pruning methods*** [1-5] and (2) ***general understanding of the encoder-decoder model architecture which is still controversial*** [6,7]. The significance of our proposed method based on such observations is as follows:
>
> *Firstly*, we believe that reducing the number of decoder layers is not a generally accepted starting point for network pruning. This is because, as we mentioned before, the efficient architectures of encoder-decoder models are still controversial topics. Moreover, before our studies, people could not guarantee to maintain the model performance by using depth pruning. The detailed description is below (see * in detailed description).
>
> Secondly, it is naturally desired to reduce the number of parameters as long as the performance does not deteriorate. Although the structured pruning on the encoder network can cause severe performance degradation (see Figure 2 and Section 3), our well-constructed width pruning scheme generally has three advantages: (1) increasing model performance, (2) slightly improving inference speed, (3) enlarging memory efficiency. While the first two advantages were depicted in our submission (see Section 5.3 and Table 3), we further discussed them in detail below (see ** in detailed description).
>
> **Detailed Description:**
>
> Here, we emphasize the contribution of our proposed method in detail.
>
> (*) There are two reasons why we believe starting with pruning the number of layers in the decoder is unnatural:
>
> * ***Firstly, the efficient architectures of encoder-decoder models are still controversial topics.*** While our discovery on the number of decoder layers somewhat consistent with traditional methods [6], this discovery can be shown as not only revisiting the superiority of such a traditional approach which is conflicts with more recent argument [7] that are deep and narrow decoder network is computationally efficient and achieves higher performance. Furthermore, our provided findings also suggest a more general perspective which holds on a wider range of NLP tasks (i.e., both NLG and NLU tasks).
>
> * ***Secondly, before our studies, people could not guarantee to maintain the model performance by using depth pruning.*** In the recent research direction, most structured pruning methods [1-5] exploit fine-grained pruning rather than layer-dropping method [8,9] which can cause a larger performance drop. However, we systematically show that depth pruning method is quite usable in the decoder network of encoder-decoder models that is counter-intuitive to recent studies.
>
> (**) Moreover, we believe that our width pruning on the encoder network is also important, as we mentioned above. Our contributions are from three factors:
>
> * Increasing model performance: As depicted in Figure 2 and Section 3, the sparsity of encoder network is the dominant factor for model performance, and simply conducting SOTA structured pruning method [3] can cause severe performance degradation. However, as we can see from Table 3 in our paper, our proposed approach slightly improves the performance compared to without width pruning. Furthermore, we prevent severe performance degradation from naïvely using pruning on the encoder network by guiding to control encoder sparsity which has a significant impact on overall model performance.
>
> * Improving inference speed: As we can see from Table 3 in our paper, our proposed approach also improves the inference speed by about ***5~10%*** compared to w/o encoder pruning as well as increases the model performance. Furthermore, we observed that this ***relative speed gap becomes even more pronounced, increasing up to 15-30%, when we conduct experiments on a CPU*** (see below Table). This is because of the difference in effectiveness for parallel computation between GPU and CPU. However, our overall method still shows the computational efficiency as the proportion of decoder computation is still larger than encoder computation. We believe that this improvement is not insignificant.
>
> >> |                    | SAMSum |       | TweetQA |       |
> >> |--------------------|:------:|:-----:|:-------:|:-----:|
> >> | T5-Base            |  43.33 | ×1.00 |  57.16  | ×1.00 |
> >> | NASH (w/o L0 Reg.) |  41.06 | ×2.50 |  54.73  | ×1.89 |
> >> | NASH (w/ L0 Reg.)  |  41.4  | ×2.86 |  55.08  | ×2.47 |
>
> * Enlarge memory efficiency: It is naturally desirable to reduce as many parameters as possible without raising performance degradation. Combined with depth pruning on decoder network, the proportion of encoder parameters will be increased. As a result, we can achieve relatively high memory efficiency with low encoder sparsity. For example, in the above Table, the numbers of Transformer layers parameters are 112.6M and 88.7M for w/o and w/ L0 regularization, which means that our narrow encoder strategy has higher memory efficiency (***can save about 20% memory*** compared to 10% in w/o depth pruning of decoder network).
>
> ***Q2. Possibility of making general purpose language models rather than task-specific language models***
>
> ***A2.***
> To show the possibility that our proposed method NASH can be used in making general purpose language models, we conducted experiments on two complex scenarios: (1) multi-task scenario (2) instruction-tuning scenario. We first describe our experimental setup and then describe the experimental results.
>
> **(1) Multi-task scenario:**
>  We consider the multi-task scenario that includes seven tasks (BoolQ, TweetQA, SAMSum, CoLA, STS-B, SST-2, WiC). We applied our method on multiple downstream tasks in the training stage, and evaluated the model on each individual task. Following settings of generation tasks, we set sparsity ratio as 30% for encoder, and set 2, 3, and 4 layers for decoder layer. NASH with 2, 3, or 4 layers outperforms T5-small in terms of both generation performance and inference speed.
>
> |          | BoolQ (Acc) | (Speedup) | TweetQA (Meteor) | (Speedup) | SAMSum (Rouge-L) | (Speedup) | CoLA (Mcc) | (Speedup) | STS-B (Corr) |(Speedup) | SST-2 (Acc) | (Speedup) | WiC (Acc) | (Speedup) |
> |----------|:---------------:|:-----:|:--------------------:|:-----:|:--------------------:|:-----:|:--------------:|:-----:|:----------------:|:-----:|:---------------:|:-----:|:-------------:|:-----:|
> | T5-base  |      77.03      | ×1.00 |         57.65        | ×1.00 |         40.93        | ×1.00 |      52.56     | ×1.00 |       87.86      | ×1.00 |      92.78      | ×1.00 |     66.14     | ×1.00 |
> | T5-small |      72.02      | ×1.98 |         50.45        | ×2.06 |         39.50        | ×2.32 |      20.52     | ×2.14 |       86.05      | ×2.06 |      91.51      | ×2.71 |     65.20     | ×2.64 |
> | NASH 2   |      76.82      | ×2.66 |         51.69        | ×4.67 |         39.02        | ×5.36 |      43.38     | ×3.46 |       86.79      | ×3.15 |      91.63      | ×2.71 |     65.20     | ×2.64 |
> | NASH 3   |      76.94      | ×2.28 |         52.21        | ×3.46 |         39.85        | ×4.14 |      47.13     | ×2.73 |       87.67      | ×2.57 |      91.86      | ×2.38 |     66.46     | ×2.33 |
> | NASH 4   |      77.40      | ×2.03 |         53.70        | ×2.81 |         40.40        | ×3.30 |      50.36     | ×2.30 |       87.51      | ×2.21 |      92.66      | ×2.08 |     65.52     | ×2.02 |
>
> **(2) Instruction-tuning:**
> We consider the instruction-tuning scenario with the databricks-dolly-15k dataset. We split the total dataset into 14k of train samples,1k of evaluation samples. We evaluate the trained model (with 14 of train samples) on three instruction-following datasets to check the generalizability of model across the different datasets: (1) 1k dolly evaluation samples; (2) SelfInst [10]; and (3) Vicuna evaluation [11] datasets.
>
> |          | Dolly Eval (ROUGE-L) | (ms/example)  | (speedup) | SelfInst (ROUGE-L) | (ms/example)  | (speedup) | Vicuna Eval (ROUGE-L) | (ms/example)  | (speedup) |
> |----------|:----------:|:------:|:-----:|:--------:|:------:|:-----:|:-----------:|:------:|:-----:|
> | T5-base  |    32.53   | 104.57 | ×1.00 |   14.20  | 113.39 | ×1.00 |    20.47    | 146.52 | ×1.00 |
> | T5-small |    29.29   | 53.408 | ×1.96 |   11.67  |  59.19 | ×1.91 |    17.40    |  66.39 | ×2.21 |
> | NASH 2   |    28.09   | 20.307 | ×5.15 |   10.58  |  21.50 | ×5.27 |    20.09    |  24.83 | ×5.90 |
> | NASH 3   |    30.87   | 28.856 | ×3.62 |   11.91  |  29.92 | ×3.79 |    21.83    |  34.51 | ×4.24 |
> | NASH 4   |    31.43   | 39.212 | ×2.67 |   13.01  |  38.53 | ×2.94 |    21.07    |  46.29 | ×3.17 |
>
> Through the results shown in above scenarios, we believe that our proposed NASH can be utilized to make more efficient general purpose language models.
>
> ***References***
>
> [1] Structured pruning of large language models. EMNLP 2020
> [2] Block Pruning For Faster Transformers. EMNLP. 2021
> [3] Structured Pruning Learns Compact and Accurate Models. ACL. 2022
> [4] A Fast Post-Training Pruning Framework for Transformers. NeurIPS. 2022
> [5] What Matters In The Structured Pruning of Generative Language Models?. ArXiv. 2023
> [6] Deep Encoder, Shallow Decoder: Reevaluating Non-autoregressive Machine Translation. ICLR. 2021
> [7] Scale Efficiently: Insights from Pre-training and Fine-tuning Transformers. ICLR. 2022
> [8] Reducing Transformer Depth on Demand with Structured Dropout. ICLR. 2020
> [9] On the Effect of Dropping Layers of Pre-trained Transformer Models. ArXiv. 2020
> [10] Self-Instruct: Aligning Language Models with Self-Generated Instructions. ACL. 2023
> [11] Vicuna: An Open-Source Chatbot Impressing GPT-4 with 90%* ChatGPT Quality. 2023

---

### Official Review · Reviewer_1C5A · 2023-08-05

**Soundness:** 4

**Excitement:**

4: Strong: This paper deepens the understanding of some phenomenon or lowers the barriers to an existing research direction.

**Paper Topic And Main Contributions:**

The proposed paper introduces NASH, a novel method to structural pruning for encoder-decoder models. NASH shows superior performance on many NLU and NLG tasks. The paper generalizes some previous insights on training of encoder-decoder models, e.g. that a rule of thumb is "deep encoder, shallow decoder".

**Questions For The Authors:**

1. How the proposed method correspond with structural pruning for decoder-only models?

**Reasons To Accept:**

Although model pruning is not my primary field, I have to admit that reading the paper was quite smooth. I appreciate the split of preliminary experiments and the methods, since it helps to understand why some decision were made. The evaluation seems to be done properly.

The paper is well written.

**Reasons To Reject:**

The experiments are conducted only using T5 architecture, it would make sense to make ablation on any non-T5 model, for example BART or vanilla transformer.



**Reproducibility:**

4: Could mostly reproduce the results, but there may be some variation because of sample variance or minor variations in their interpretation of the protocol or method.

**Reviewer Confidence:**

1: Not my area, or paper was hard for me to understand. My evaluation is just an educated guess.

---

> ### Author Rebuttal · Authors · 2023-08-29
>
> We are grateful that you have acknowledged our well-organized paper and well-constructive experiments. The answers for your concerns are addressed as follows.
>
> ***Q1. Experiments conducted on any non-T5 model, for example BART or vanilla Transformer.***
>
> **A1.** We conducted experiments on T5 models because they are more popular and widely used across various tasks, such as code generation, medical text understanding, and spell checking [1-4] than other encoder-decoder models like BART [5] and PEGASUS [6].
>
>
> To address your concern, we conducted supplementary experiments employing another encoder-decoder model, BART, on the SAMSum dataset. Specifically, we compare NASH-BART-large (conducting NASH on BART-large), and BART-base models.
> As illustrated in the table below, our results with the BART demonstrate consistent performance when compared to the T5 case. Notably, BART NASH 2 exhibits an impressive x5.6 faster inference speed than BART-large, while BART NASH 3 achieves faster inference compared to BART-base, all while maintaining similar output quality.
>
> |         |  BART |        | NASH-BART|           |          |
> |:-------:|:-----:|:------:|:--------:|:---------:|:--------:|
> |         | large |  base  | 2 layers |  3 layers | 4 layers |
> | ROUGE-L | 43.24 |  42.39 |   40.47  |   42.33   |   42.85  |
> | Speedup | ×1.00 | ×2.05  |  ×5.56   |   ×3.38   |  ×2.43   |
>
> ***Q2. How does the proposed method correspond with structured pruning for decoder-only models?***
>
> **A2.**
> In this work, we discussed that the well-performing structured pruning method method for the encoder-only model [7] does not work well for the encoder-decoder models, and optimized the structured pruning method for the encoder-decoder models. Similarly, we believe that further discussion is needed to find a well-functioning structured pruning method for decoder-only models.
>
> In response to your question, when comparing decoder-only models to encoder-decoder models, the latter can be segmented into two parts. It's noteworthy that these two networks exhibit contrasting properties, as illustrated in Figure 2: while (1) the performances of LMs are vulnerable to the number of encoder layers, but robust to the number of decoder layers, (2) the number of the decoder layers is primary factor of inference speed of the LMs, but the number of the encoder layers is insignificant to inference speedup. Our proposed method fully exploits such property by conducting different pruning strategies on encoder and decoder networks. This allows us to use a maximally 6x smaller decoder network without severe performance degradation but achieving up to 5x faster inference speed.
>
> However, directly applying our proposed method to decoder-only models could be impossible or cause severe performance degradation, because such models process both encoding input context and generation outputs unlike encoder-decoder models. One suggested modification is (1) slightly conducting width pruning with moderate sparsity on the language model, similar to our narrow encoder (Section 4.2), then (2) only using layer skipping for auto-regressive generation process while fully utilizing the whole network for processing input context. We believe that such modification could be working on decoder-only models.
>
> ***References***
>
> [1] ClinicalT5: A Generative Language Model for Clinical Text. Findings of EMNLP. 2022
> [2] CodeT5: Identifier-aware Unified Pre-trained Encoder-Decoder Models for Code Understanding and Generation. EMNLP. 2021
> [3] CodeT5+: Open Code Large Language Models for Code Understanding and Generation. ArXiv. 2023
> [4] ByT5: Towards a Token-Free Future with Pre-trained Byte-to-Byte Models. TACL. 2022
> [5] BART: Denoising Sequence-to-Sequence Pre-training for Natural Language Generation, Translation, and Comprehension. ACL. 2020
> [6] PEGASUS: Pre-training with Extracted Gap-sentences for Abstractive Summarization. ICML. 2020
> [7] Structured Pruning Learns Compact and Accurate Models. ACL. 2022

---

### Official Review · Reviewer_h8AR · 2023-08-09

**Soundness:** 4

**Excitement:**

2: Mediocre: This paper makes marginal contributions (vs non-contemporaneous work), so I would rather not see it in the conference.

**Missing References:**

Michel, Paul, Omer Levy, and Graham Neubig. "Are sixteen heads really better than one?." Advances in neural information processing systems 32 (2019).

**Paper Topic And Main Contributions:**

In this work, the authors provide a rigorous study to explore the dominant factors that trade-off performance and efficiency during structured pruning. The key observations are that reducing the decoder layers is the best option for acceleration while an appropriate sparsity imposed on the encoder slightly improves the quality. Based on these observations, the authors propose NASH, a narrow encoder and shallow decoder pruning strategy. Extensive experiments have demonstrated the effectiveness of NASH. In general, this paper is well-written and I am satisfied with the comprehensive experiments and studies conducted by the authors. However, I find that the conclusion and method are somewhat trivial. For example, the contribution of the decoder layer to efficiency and quality in encoder-decoder models has long been observed, as also pointed out in the paper. I think the observations in the first half of this paper are expected and not very interesting. The proposed method is also straightforward: it looks to me like a direct application of L0 regularization on encoder-decoder models with a targeted granularity (layer for decoder and matrix for encoder).

**Questions For The Authors:**

1. Why do the authors choose L0 regularization to instantiate their method? Does it provide any advantage? How about the gradient-based pruning methods?

**Reasons To Accept:**

1. Rigorous analysis and extensive experiments.
2. A simple and effective approach.

**Reasons To Reject:**

1. The observations are somewhat trivial and obvious given the previous work.

**Reproducibility:**

5: Could easily reproduce the results.

**Reviewer Confidence:**

3: Pretty sure, but there's a chance I missed something. Although I have a good feel for this area in general, I did not carefully check the paper's details, e.g., the math, experimental design, or novelty.

---

> ### Author Rebuttal · Authors · 2023-08-29
>
> We thank you for your constructive comments, and we are encouraged that you pointed out the simplicity and effectiveness of our proposed method based on our rigorous analysis. The answers for your concerns are addressed as follows.
>
> ***Q1. The main contributions of observations in Section 3***
>
> **A1.**
> **Summary:**
>
> The observations in our paper provide (1) the ***counter-intuitive perspective on the recent direction of structured pruning methods*** [1-5] ***(see First in detailed description)*** and (2) ***general understanding to the encoder-decoder model architecture which is still controversial*** [8,9] ***(see Second in detailed description)***. Hence, they are not trivial and obvious, rather valuable and explainable. Moreover, we provide exciting results on the multi-task and instruction-tuning datasets, that our proposed method can be possible to apply for making general language models. We expect such additional results to intrigue you. The detailed descriptions are demonstrated below.
>
> **Detailed Descriptions:**
>
> *Firstly*, while the recent studies [1-5] showed fine-grained structured pruning (prune individual head in MHA or individual dimension in FFN) [1-4] can achieve both higher model performance and inference speedup than the traditional layer-dropping based methods [6, 7], we empirically showed that the layer-dropping methods can achieve competitive performances as fine-grained structured pruning [3], while highly increasing inference speed.
>
> *Secondly*, while our discovery of the number of decoder layers somewhat consistent with traditional methods [8], this discovery can be shown as not only revisiting the superiority of such a traditional approach which is conflicts with more recent argument [9] that are deep and narrow decoder network is computationally efficient and achieves higher performance, but also suggest more general perspective. We also figured out that the moderate level of sparsity imposed on the encoder slightly improves the quality although it is memory-efficient.
>
> To further demonstrate that our findings are applicable to various settings, we conducted experiments on two complex scenarios: (1) multi-task scenario (2) instruction-tuning scenario. We first describe our experimental setup and then describe the experimental results.
>
> **(1) Multi-task scenario:**
>  We consider the multi-task scenario that includes seven tasks (BoolQ, TweetQA, SAMSum, CoLA, STS-B, SST-2, WiC). We applied our method on multiple downstream tasks in the training stage, and evaluated the model on each individual task. Following settings of generation tasks, we set sparsity ratio as 30% for encoder, and set 2, 3, and 4 layers for decoder layer. NASH with 2, 3, or 4 layers outperforms T5-small in terms of both generation performance and inference speed.
>
> |          | BoolQ (Acc) | (Speedup) | TweetQA (Meteor) | (Speedup) | SAMSum (Rouge-L) | (Speedup) | CoLA (Mcc) | (Speedup) | STS-B (Corr) |(Speedup) | SST-2 (Acc) | (Speedup) | WiC (Acc) | (Speedup) |
> |----------|:---------------:|:-----:|:--------------------:|:-----:|:--------------------:|:-----:|:--------------:|:-----:|:----------------:|:-----:|:---------------:|:-----:|:-------------:|:-----:|
> | T5-base  |      77.03      | ×1.00 |         57.65        | ×1.00 |         40.93        | ×1.00 |      52.56     | ×1.00 |       87.86      | ×1.00 |      92.78      | ×1.00 |     66.14     | ×1.00 |
> | T5-small |      72.02      | ×1.98 |         50.45        | ×2.06 |         39.50        | ×2.32 |      20.52     | ×2.14 |       86.05      | ×2.06 |      91.51      | ×2.71 |     65.20     | ×2.64 |
> | NASH 2   |      76.82      | ×2.66 |         51.69        | ×4.67 |         39.02        | ×5.36 |      43.38     | ×3.46 |       86.79      | ×3.15 |      91.63      | ×2.71 |     65.20     | ×2.64 |
> | NASH 3   |      76.94      | ×2.28 |         52.21        | ×3.46 |         39.85        | ×4.14 |      47.13     | ×2.73 |       87.67      | ×2.57 |      91.86      | ×2.38 |     66.46     | ×2.33 |
> | NASH 4   |      77.40      | ×2.03 |         53.70        | ×2.81 |         40.40        | ×3.30 |      50.36     | ×2.30 |       87.51      | ×2.21 |      92.66      | ×2.08 |     65.52     | ×2.02 |
>
> **(2) Instruction-tuning:**
> We consider the instruction-tuning scenario with the databricks-dolly-15k dataset. We split the total dataset into 14k of train samples,1k of evaluation samples. We evaluate the trained model (with 14 of train samples) on three instruction-following datasets to check the generalizability of model across the different datasets: (1) 1k dolly evaluation samples; (2) SelfInst [10]; and (3) Vicuna evaluation [11] datasets.
>
> |          | Dolly Eval (ROUGE-L) | (ms/example)  | (speedup) | SelfInst (ROUGE-L) | (ms/example)  | (speedup) | Vicuna Eval (ROUGE-L) | (ms/example)  | (speedup) |
> |----------|:----------:|:------:|:-----:|:--------:|:------:|:-----:|:-----------:|:------:|:-----:|
> | T5-base  |    32.53   | 104.57 | ×1.00 |   14.20  | 113.39 | ×1.00 |    20.47    | 146.52 | ×1.00 |
> | T5-small |    29.29   | 53.408 | ×1.96 |   11.67  |  59.19 | ×1.91 |    17.40    |  66.39 | ×2.21 |
> | NASH 2   |    28.09   | 20.307 | ×5.15 |   10.58  |  21.50 | ×5.27 |    20.09    |  24.83 | ×5.90 |
> | NASH 3   |    30.87   | 28.856 | ×3.62 |   11.91  |  29.92 | ×3.79 |    21.83    |  34.51 | ×4.24 |
> | NASH 4   |    31.43   | 39.212 | ×2.67 |   13.01  |  38.53 | ×2.94 |    21.07    |  46.29 | ×3.17 |
>
> Through the results shown in above complex scenarios, we strongly argue that the generation performance of encoder-decoder models is quite robust to the number of decoder layers. These results contradict conventional insights and disprove that our findings are not trivial or obvious, but rather very surprising and valuable.
>
> ***Q2-1. The advantage of L0 regularization***
>
> **A2-1.** Most of the research in structured pruning domains are mainly conducted experiments on encoder-only models such as BERT or RoBERTa. Among the various approaches in the research direction of encoder-only structured pruning method, recent L0 regularization method has mainly two advantages: Firstly, L0 regularization is easier to control the sparsity precisely. [1,3] We also find that such methods controls well in the low sparsity regime of encoder networks, although it did not work well on severe sparsity of the encoder or decoder networks. Secondly, In the recent work of L0 regularization, CoFi [3], achieved SoTA performance compared to gradient-based method [2,12] or layer-dropping method [6,7]. To exploit such an advantage of L0 regularization shown in encoder-only models, we chose L0 regularization to instantiate our proposed method, NASH.
>
> ***Q2-2. Comparison with gradient-based pruning methods?***
>
> **A2-2.** To address your question, we conducted experiments where we replaced our gradual L0 regularization with a gradient-based pruning method. Following [12-15], we compute the importance score of individual attention head and neuron in FFN by using first-order Taylor expansion of gradient information. Due to the less flexibility of gradient-based pruning, we uniformly prune the individual heads (4 heads for all layers) and neurons (~30% dimension of FFN) for all Transformer layers.
>
> |                   |  RTE (Acc) | (Speedup) | SAMSum (ROUGE-L) | (Speedup) |
> |-------------------|:-----:|:-----:|:------:|:-----:|
> | Gradient-based    | 76.17 | ×2.27 |  40.85 | ×2.97 |
> | L0 Regularization | 78.70 | ×2.24 |  41.34 | ×2.99 |
>
> The results show that NASH with L0 regularization can achieve higher performance than the gradient-based pruning method in both NLU (RTE) and NLG (SAMSum) tasks.
>
> ***Q3. Missing reference***
>
> **A3.** Thank you for pointing out the missing reference [16]. We will add it to our revised version.
>
> ***References***
>
> [1] Structured pruning of large language models. EMNLP 2020
> [2] Block Pruning For Faster Transformers. EMNLP. 2021
> [3] Structured Pruning Learns Compact and Accurate Models. ACL. 2022
> [4] A Fast Post-Training Pruning Framework for Transformers. NeurIPS. 2022
> [5] What Matters In The Structured Pruning of Generative Language Models?. ArXiv. 2023
> [6] Reducing Transformer Depth on Demand with Structured Dropout. ICLR. 2020
> [7] On the Effect of Dropping Layers of Pre-trained Transformer Models. ArXiv. 2020
> [8] Deep Encoder, Shallow Decoder: Reevaluating Non-autoregressive Machine Translation. ICLR. 2021
> [9] Scale Efficiently: Insights from Pre-training and Fine-tuning Transformers. ICLR. 2022
> [10] Self-Instruct: Aligning Language Models with Self-Generated Instructions. ACL. 2023
> [11] Vicuna: An Open-Source Chatbot Impressing GPT-4 with 90%* ChatGPT Quality. 2023
> [12] DynaBERT: Dynamic BERT with Adaptive Width and Depth. NeurIPS. 2020
> [13] Pruning convolutional neural networks for resource efficient inference. NIPS. 2017
> [14] Analyzing multi-head self-attention: Specialized heads do the heavy lifting, the rest can be pruned. ACL. 2019
> [15] LLM-Pruner: On the Structural Pruning of Large Language Models. ArXiv. 2023
> [16] Are sixteen heads really better than one?. NeurIPS. 2019

---

### Official Review · Reviewer_7MBx · 2023-08-11

**Soundness:** 3

**Excitement:**

3: Ambivalent: It has merits (e.g., it reports state-of-the-art results, the idea is nice), but there are key weaknesses (e.g., it describes incremental work), and it can significantly benefit from another round of revision. However, I won't object to accepting it if my co-reviewers champion it.

**Paper Topic And Main Contributions:**

This study explores impact of structured pruning on encoder-decoder models, revealing the importance of decoder layers for inference speed and the role of sparse pruning in enhancing encoder network quality. Building on these insights, the NASH framework is introduced, effectively refining encoder and decoder networks. Experiments across diverse tasks confirm NASH's success in improving both computational efficiency and output quality.

**Questions For The Authors:**

The Sec."Preliminary," covers Transformers and Structured Pruning, both of which discuss the work of others. In what aspects does the author's improvement primarily manifest across these two parts?

**Reasons To Accept:**

The paper is well-written and easy to follow. The illustration of motivation and method is clear. The ablation
study is reasonably thorough and acceptable results are claimed.

**Reasons To Reject:**

1.The author concludes that the number of decoder layers is the primary factor for accelerating inference speed. The depth of the decoder affects the model's complexity and capability, as deeper decoders can generally capture more intricate semantics and relationships, thereby enhancing the quality of tasks such as generation and translation. The author should quantitatively assess and present experimental results regarding semantic relationships in complex scenarios. Additionally, the number of encoder layers can also impact the inference speed to varying degrees, yet the author fails to analyze why this aspect has been excluded.
2.The math presentation can be significantly improved. I think to add a notation table can help.
3.Compared to the SOTA methods, the proposed approach showcases its advantages in various aspects, such as reduced computational complexity and enhanced efficiency. The article should include charts and graphs to visually depict the method's strengths, including its specific computational complexity.
4.The author should incorporate more comparative results with SOTA methods in the experimental tables.
5.The author lacks a qualitative analysis of the feasibility of this approach considering factors such as task complexity, data volume, real-time requirements, and model robustness.

**Reproducibility:**

4: Could mostly reproduce the results, but there may be some variation because of sample variance or minor variations in their interpretation of the protocol or method.

**Reviewer Confidence:**

4: Quite sure. I tried to check the important points carefully. It's unlikely, though conceivable, that I missed something that should affect my ratings.

---

> ### Author Rebuttal · Authors · 2023-08-29
>
> We thank you for your thoughtful feedback. We are encouraged that you found our paper well-organized, clear motivation and methods, and our experimental results comprehensive. The answers for your concerns are addressed as follows.
>
> ***Q1-1. Evaluation of shallow decoder on complex scenarios.***
>
> **A1-1**.
> **Summary:**
>
> As described in Figure 2 and Section 3, we figured out that the most important factor for the generation quality is the sparsity of the encoder network, and the number of the decoder layers is relatively unimportant.
>
> To address your question, we additionally conduct experiments on two complex scenarios: (1) multi-task scenario with seven tasks and (2) instruction-tuning task. ***The results on such complex scenarios demonstrate that our finding does not originate from the low difficulty of target tasks, but validates generally.*** The detailed experimental setup and results are below.
>
> **Detailed Results:**
>
> **(1) Multi-task scenario:**
>  We consider the multi-task scenario that includes seven tasks (BoolQ, TweetQA, SAMSum, CoLA, STS-B, SST-2, WiC). We applied our method on multiple downstream tasks in the training stage, and evaluated the model on each individual task. Following settings of generation tasks, we set sparsity ratio as 30% for encoder, and set 2, 3, and 4 layers for decoder layer. NASH with 2, 3, or 4 layers outperforms T5-small in terms of both generation performance and inference speed.
>
> |          | BoolQ (Acc) | (Speedup) | TweetQA (Meteor) | (Speedup) | SAMSum (Rouge-L) | (Speedup) | CoLA (Mcc) | (Speedup) | STS-B (Corr) |(Speedup) | SST-2 (Acc) | (Speedup) | WiC (Acc) | (Speedup) |
> |----------|:---------------:|:-----:|:--------------------:|:-----:|:--------------------:|:-----:|:--------------:|:-----:|:----------------:|:-----:|:---------------:|:-----:|:-------------:|:-----:|
> | T5-base  |      77.03      | ×1.00 |         57.65        | ×1.00 |         40.93        | ×1.00 |      52.56     | ×1.00 |       87.86      | ×1.00 |      92.78      | ×1.00 |     66.14     | ×1.00 |
> | T5-small |      72.02      | ×1.98 |         50.45        | ×2.06 |         39.50        | ×2.32 |      20.52     | ×2.14 |       86.05      | ×2.06 |      91.51      | ×2.71 |     65.20     | ×2.64 |
> | NASH 2   |      76.82      | ×2.66 |         51.69        | ×4.67 |         39.02        | ×5.36 |      43.38     | ×3.46 |       86.79      | ×3.15 |      91.63      | ×2.71 |     65.20     | ×2.64 |
> | NASH 3   |      76.94      | ×2.28 |         52.21        | ×3.46 |         39.85        | ×4.14 |      47.13     | ×2.73 |       87.67      | ×2.57 |      91.86      | ×2.38 |     66.46     | ×2.33 |
> | NASH 4   |      77.40      | ×2.03 |         53.70        | ×2.81 |         40.40        | ×3.30 |      50.36     | ×2.30 |       87.51      | ×2.21 |      92.66      | ×2.08 |     65.52     | ×2.02 |
>
> **(2) Instruction-tuning:**
>  We consider the instruction-tuning scenario with the databricks-dolly-15k dataset. We split the total dataset into 14k of train samples,1k of evaluation samples. We evaluate the trained model (with 14 of train samples) on three instruction-following datasets to check the generalizability of model across the different datasets: (1) 1k dolly evaluation samples; (2) SelfInst [1]; and (3) Vicuna evaluation [2] datasets.
>
> |          | Dolly Eval (ROUGE-L) | (ms/example)  | (speedup) | SelfInst (ROUGE-L) | (ms/example)  | (speedup) | Vicuna Eval (ROUGE-L) | (ms/example)  | (speedup) |
> |----------|:----------:|:------:|:-----:|:--------:|:------:|:-----:|:-----------:|:------:|:-----:|
> | T5-base  |    32.53   | 104.57 | ×1.00 |   14.20  | 113.39 | ×1.00 |    20.47    | 146.52 | ×1.00 |
> | T5-small |    29.29   | 53.408 | ×1.96 |   11.67  |  59.19 | ×1.91 |    17.40    |  66.39 | ×2.21 |
> | NASH 2   |    28.09   | 20.307 | ×5.15 |   10.58  |  21.50 | ×5.27 |    20.09    |  24.83 | ×5.90 |
> | NASH 3   |    30.87   | 28.856 | ×3.62 |   11.91  |  29.92 | ×3.79 |    21.83    |  34.51 | ×4.24 |
> | NASH 4   |    31.43   | 39.212 | ×2.67 |   13.01  |  38.53 | ×2.94 |    21.07    |  46.29 | ×3.17 |
>
> Through the results shown in above complex scenarios, we strongly argue that the generation performance of encoder-decoder models is quite robust to the number of decoder layers. These results contradict conventional insights and disprove that our findings are not trivial or obvious, but rather very surprising and valuable.
>
> ***Q1-2. Analysis for the impact of the number of encoder layers on inference speed***
>
> **A1-2.**
> **Summary:**
>
> As shown in Figure 2-4 and Section 3, we discussed the impact of the number of encoder layers as a less important factor based on our experimental results. The main reasons that we referred but did not emphasize the relationship between the number of encoder layers and the inference speedup are twofold: Firstly, depth pruning on the encoder networks is not effective for the inference speedup (see Figure 2, 4); Secondly, applying high-sparsity depth pruning to the encoder networks is impractical, given the crucial role played by the encoder network in maintaining performance (see Figure 2 and Finding 3.2). Considering the trade-off between small advantage (inference speedup) and the enormous disadvantage (performance degradation), we decided to exclude the high-sparsity depth pruning on the encoder network. More detailed description are below:
>
> **Detailed description:**
>
> *Firstly*, as depicted in Figure 2, although we conducted pruning on the encoder networks with sparsity up to 95%, the maximal speedup gain was ×1.3. This is because, the most of processing time is spent on the autoregressive generation process as summarized in Figure 4, even though we conducted experiments on NLU tasks which have relatively larger proportional of encoder processing time.
>
> *Secondly*, as shown in the first row of Figure 2, the sparsity of the encoder network is the primary factor for maintaining the output quality of encoder-decoder models, hence, we cannot enlarge the sparsity of encoder networks. In contrast to the recent literature concerning encoder-only models [3], it is surprising that the encoder network of encoder-decoder models are vulnerable to the pruning even though the sparsity is not severely high.
>
> To address this phenomenon, we conjecture that the generation performance of the encoder-decoder model depends largely upon the encoded context rather than the depth of the decoder network. The similar trends discovered in early-exiting frameworks [4,5] that only add the multiple classifiers in intermediate decoder layers.
>
> Moreover, when the sparsity is moderate (\~60% sparsity), we observe that the difference between pruning encoder w/ and w/o layer are negligible in terms of both inference speed and generation quality. This result is consistent with the previous study [3] in the structured pruning on the encoder-only model that “in the moderate regime, the number of the layers does not affect the model performance and inference speedup”. Although there is a small amount of difference in the high sparsity regimes (60~95%), we can say that such difference is meaningless because of their low feasibility and useless generation performances.
>
> ***Q2. Suggestion for adding a notation table.***
>
> **A2.** Thank you for your constructive comment. To improve the clarity, we will add a following notation table in the finalized version.
>
> |        Type       |      Notation      |                                Description                               |
> |:-----------------:|:------------------:|:------------------------------------------------------------------------:|
> |  Layer selection  |       $L_{s}$      |                      a set of selected layer indices                     |
> |                   |       $d_{s}$      |        the number of selected layers for the decoder depth pruning       |
> |  Model components |  $H_{dec, s}^{l}$  | a hidden representation of the $l$ th student decoder layer              |
> | L0 regularization | $z_{head}^{(i,j)}$ | a mask for a $j$ th attention head of an $i$ th layer                    |
> |                   | $z_{int}^{ (i,j)}$ | a mask for a $j$ th dimension of feed-forward network of an $i$ th layer |
> |                   |      $\hat s$      |  current sparsity                                                        |
> |                   |         $t$        |  the target sparsity we set                                              |
> |                   |    $\mathcal{R}$   |  the L0 regularization loss (pruning loss)                               |
>
> ***Q3. Request for absolute inference time (specific computational complexity).***
>
> **A3.** Thank you for your constructive comment. As it is not possible to post a chart or graph in OpenReview, we post a table reporting the absolute values of inference time. We reported the absolute inference time for T5 (base, small for Figure 5, 7 and mini for Figure 5), CoFi-T5 with sparsity of 60%-90%, and NASH-T5 with decoder layers of 2-8. We use 30% of encoder sparsity on NASH.
>
> - Absolute values for Figure 5 (milliseconds/example)
>
> |               | Original T5 |       |      | CoFi-T5 (Sparsity) |      |       |       | NASH-T5 (# decoder layers) |      |      |      |      |
> |---------------|:-----------:|:-----:|:----:|:------------------:|:----:|:-----:|:-----:|:--------------------------:|:----:|:----:|:----:|:----:|
> |               |     base    | small | mini |         90%        |  80% |  70%  |  60%  |              2             |   3  |   4  |   6  |   8  |
> | TweetQA       |     9.93    |  5.16 | 3.68 |        6.25        | 7.00 |  8.75 |  8.93 |            2.47            | 3.33 | 3.94 | 5.42 | 8.66 |
> | XSum          |     41.1    |  21.0 | 17.2 |        34.0        | 36.8 |  37.4 |  40.3 |            11.0            | 17.0 | 25.1 | 30.1 | 36.0 |
> | SAMSum        |     62.4    |  29.2 | 22.9 |        43.0        | 50.7 |  54.3 |  61.8 |            11.3            | 14.7 | 20.9 | 32.2 | 42.1 |
> | CNN/DailyMail |    105.7    |  50.1 | 36.2 |        65.1        | 84.4 | 102.6 | 104.6 |            21.5            | 31.3 | 39.3 | 59.7 | 78.5 |
>
>
> - Absolute values for Figure 7 (milliseconds/example)
>
> |       | Original T5 |       | CoFi-T5 (Sparsity)     |  |      |      | NASH-T5 (# decoder layers)     |      |  |      |      |
> |:-----:|:-----------:|:-----:|:----:|:------------------:|:----:|:----:|:----:|:----:|:--------------------------:|:----:|:----:|
> |       |     base    | small |  90% |         80%        |  70% |  60% |   2  |   3  |              4             |   6  |   8  |
> | CoLA  |     4.61    |  1.84 | 2.30 |        2.66        | 3.14 | 4.14 | 1.54 | 2.26 |            2.72            | 3.64 | 4.28 |
> | RTE   |     4.52    |  2.00 | 2.10 |        2.56        | 3.18 | 4.15 | 1.54 | 2.26 |            2.67            | 3.31 | 4.38 |
> | BoolQ |     3.79    |  1.48 | 1.75 |        2.33        | 2.71 | 3.26 | 1.40 | 1.81 |            2.19            | 2.63 | 3.51 |
> | CB    |     4.59    |  1.77 | 2.13 |        2.83        | 3.30 | 3.97 | 1.70 | 2.21 |            2.67            | 3.20 | 4.28 |
>
> In the final version, we would like to depict the above values on graphs.
>
> ***Q4. Additional comparison with recent SOTA.***
>
> Previously, research on the structured pruning of encoder-decoder language models was limited. Therefore, we initially compared our method to a smaller baseline model and applied the SOTA encoder pruning method to the encoder-decoder model for reference.
> However, in response to your concerns, we carried out an additional experiment. In this experiment, we applied our method to the BART-base model for the CNN/DM summarization task, directly comparing it to the method proposed in the very recent paper [6].
> As illustrated in the table below, we ensured that the model sizes of both methods are similar. In this context, our method demonstrated superior performance, surpassing the method proposed in [6], in terms of generation quality. Our NASH-BART-2 which only uses two decoder layers outperforms the method proposed in [6](with 27% sparsity), despite using smaller parameters. Moreover, it is difficult to directly compare because there is no code repository of [6], but we achieve more than three times while [6] achieves maximially 1.5 times the reported speedup on the GPU.
>
>
> | CNNDM    | BART-base | NASH-BART-2 | NASH-BART-3 | [6] (Sparsity 50%) | [6] (Sparsity 27%)  |
> |----------|-----------|-------------|-------------|--------------------|---------------------|
> | Rouge-L  | 41.68     | 40.87       | 41.14       | 39.91              | 40.39               |
> | Speedup  | ×1.00     | ×3.18       | ×2.11       | Maximally ×1.5     | Maximally ×1.5      |
> | # Params | 139M      | 80M         | 89.8M       | 70.9M              | 102.4M              |
>
>
> ***Q5. Ablation studies on the feasibility with considering factors such as task complexity, data volume, real-time requirements, and model robustness.***
>
> **A5.**
> **Summary:**
>
> To verify the feasibility of our proposed approaches, the results of ablation study we provided in our paper are: (1) task types (4 NLG tasks, 10 NLU tasks; see Table 1, Figure 5, Figure 7, Table 8); (2) model configuration robustness (feasibility on deeper decoder network; see Table 4), and (3) Comparison with efficient-T5 (see Table 9).
>
> To intrigue the reviewer 7MBx, we additionally conduct the experiments for (1) task complexity; (2) data volume; (3) absolute value of processing time. Moreover, we emphasize the results for (4) model type. All results in (1) - (4) support that our proposed NASH has a high level of feasibility. The detailed experimental setup and results are below.
>
> **Detailed Results:**
>
> **(1) Task complexity:**
>
> For the task complexity, we conducted the multi-task experiments with the number of tasks three and seven. We applied our method on multiple downstream tasks in the training stage, and evaluated the model on each individual task. All experiments were conducted on T5-base, T5-small, and NASH-T5. As we already reported the absolute inference time in A3, we only report the relative inference speedup.
>
> - Multi-task learning using 3 downstream tasks (BoolQ, TweetQA and SAMSum)
>
> - Multi-task learning using 7 downstream tasks (BoolQ, TweetQA, SAMSum, CoLA, STS-B, SST-2, WiC)
>
> As the number of tasks increases (see BoolQ, TweetQA, SAMSum), the decline in performance is very large in T5-small, while the NASH-T5-base are robust. This means that our proposed method is more usable in the complex scenario.
>
> - Multi-task learning using 3 downstream tasks (BoolQ, TweetQA, and SAMSUM)
>
> |          | BoolQ (Acc) | (Speedup) | TweetQA (Meteor) | (Speedup) | SAMSum  (Rouge-L) | (Speedup) |
> |----------|:-----------------:|:-----:|:----------------------:|:-----:|:----------------------:|:-----:|
> | T5-base  |       78.23       | ×1.00 |          58.39         | ×1.00 |          41.12         | ×1.00 |
> | T5-small |       71.41       | ×1.96 |          50.70         | ×2.00 |          39.47         | ×2.22 |
> | NASH 2   |       77.52       | ×2.65 |          53.03         | ×4.82 |          39.32         | ×4.84 |
> | NASH 3   |       77.40       | ×2.28 |          53.12         | ×3.68 |          40.59         | ×3.74 |
> | NASH 4   |       76.73       | ×1.96 |          55.99         | ×2.75 |          40.50         | ×2.93 |
>
>
> - Multi-task learning using 7 downstream tasks (BoolQ, TweetQA, SAMSUM, CoLA, STS-B, SST-w, and WiC)
>
> |          | BoolQ (Acc) | (Speedup) | TweetQA (Meteor) | (Speedup) | SAMSum (Rouge-L) | (Speedup) | CoLA (Mcc) | (Speedup) | STS-B (Corr) |(Speedup) | SST-2 (Acc) | (Speedup) | WiC (Acc) | (Speedup) |
> |----------|:---------------:|:-----:|:--------------------:|:-----:|:--------------------:|:-----:|:--------------:|:-----:|:----------------:|:-----:|:---------------:|:-----:|:-------------:|:-----:|
> | T5-base  |      77.03      | ×1.00 |         57.65        | ×1.00 |         40.93        | ×1.00 |      52.56     | ×1.00 |       87.86      | ×1.00 |      92.78      | ×1.00 |     66.14     | ×1.00 |
> | T5-small |      72.02      | ×1.98 |         50.45        | ×2.06 |         39.50        | ×2.32 |      20.52     | ×2.14 |       86.05      | ×2.06 |      91.51      | ×2.71 |     65.20     | ×2.64 |
> | NASH 2   |      76.82      | ×2.66 |         51.69        | ×4.67 |         39.02        | ×5.36 |      43.38     | ×3.46 |       86.79      | ×3.15 |      91.63      | ×2.71 |     65.20     | ×2.64 |
> | NASH 3   |      76.94      | ×2.28 |         52.21        | ×3.46 |         39.85        | ×4.14 |      47.13     | ×2.73 |       87.67      | ×2.57 |      91.86      | ×2.38 |     66.46     | ×2.33 |
> | NASH 4   |      77.40      | ×2.03 |         53.70        | ×2.81 |         40.40        | ×3.30 |      50.36     | ×2.30 |       87.51      | ×2.21 |      92.66      | ×2.08 |     65.52     | ×2.02 |
>
>
> **(2) Data volume:**
>
> For the data volume, we conducted the experiments on the SAMSum dataset with the number of data points of 1k, 5k, and 14k. Similar to the task complexity experiments, all experiments were conducted on T5-base, T5-small, and NASH-T5. Similar to above, we report the relative inference speedup.
>
> - SAMSum with 1k training samples
>
> |         | T5-base | T5-small | NASH-T5 (2 layers) | NASH-T5 (3 layers) | NASH-T5 (4 layers) |
> |:-------:|:-------:|:--------:|:------------------:|:------------------:|:------------------:|
> | ROUGE-L |  39.00  |   35.60  |        30.44       |        36.21       |        36.49       |
> | SpeedUp |  ×1.00  |   ×2.12  |        ×5.19       |        ×3.97       |        ×3.07       |
>
>
> - SAMSum with 5k training samples
>
> |         | T5-base | T5-small | NASH-T5 (2 layers) | NASH-T5 (3 layers) | NASH-T5 (4 layers) |
> |:-------:|:-------:|:--------:|:------------------:|:------------------:|:------------------:|
> | ROUGE-L |  40.55  |   38.77  |        39.41       |        41.88       |        41.89       |
> | SpeedUp |  ×1.00  |   ×2.15  |        ×5.13       |        ×4.23       |        ×3.14       |
>
> - SAMSum with 14k training samples
>
> |         | T5-base | T5-small | NASH-T5 (2 layers) | NASH-T5 (3 layers) | NASH-T5 (4 layers) |
> |--------:|:-------:|:--------:|:------------------:|:------------------:|:------------------:|
> | ROUGE-L |  43.33  |   38.14  |        38.89       |        41.09       |        41.34       |
> | SpeedUp |  ×1.00  |   ×2.14  |        ×5.53       |        ×4.24       |        ×2.99       |
>
>
> NASH-T5 beats the T5-small for all setups in terms of both model performance and inference speed, except for the model performance of NASH-T5 with two layers and 1k data points. Although our proposed method is more effective with a large number of data points, the results depict that our proposed method is quite robust to the number of data points.
>
> **(3) Absolute value of processing time:**
>
> We understand the term “real-time requirements” to mean “the absolute values of inference time”. We have reported such values in A3. Please refer to the corresponding answer.
>
> **(4) Model types:**
>
> We understand the term “model robustness” as “consistent performance according to the different model configuration”, and we have reported such values in Table 4 and Section 5.3. The results show that our proposed method NASH has robustness in terms of output performance even with shallower decoder than the original models (i.e., unpruned models) Moreover, since the original model has deeper decoder networks, NASH can achieve higher inference speedup.
>
> To address your concern, we additionally conducted experiments on BART, which is one of the encoder-decoder language models. The results demonstrate that our proposed method also works well on BART, which means the robustness to the model types.
>
> |         |  BART |        |          | NASH-BART |          |
> |:-------:|:-----:|:------:|:--------:|:---------:|:--------:|
> |         | large |  base  | 2 layers |  3 layers | 4 layers |
> | ROUGE-L | 43.24 |  42.39 |   40.47  |   42.33   |   42.85  |
> | Speedup | ×1.00 | ×2.05  |  ×5.56   |   ×3.38   |  ×2.43   |
>
>
> ***Q6. Authors’ improvement related to contents in preliminary section***
>
> **A6.** The preliminary section, we introduced the background knowledge to understand our works (or contributions) which are described in Section “Experimental Motivations”, Section “Narrow Encoder and Shallow Decoder”, and Section “Experiments”.
>
> The main contributions of our paper are: (1) While the existing structured pruning methods were mainly considered on encoder-only models [3,7,8] or decoder-only models [9], we conduct systematic analysis of structured pruning on encoder-decoder models. (2) Based on findings from our analyses, we design a simple yet effective method which fully exploits the property of different behaviors of encoder and decoder networks during structured pruning. None of these contributions were present before, which encompasses the previously mentioned works outlined in the “Preliminary” section.
>
> ***References***
>
> [1] Self-Instruct: Aligning Language Models with Self-Generated Instructions. ACL. 2023
> [2] Vicuna: An Open-Source Chatbot Impressing GPT-4 with 90%* ChatGPT Quality. 2023
> [3] Structured Pruning Learns Compact and Accurate Models. ACL. 2022
> [4] Depth-Adaptive Transformer. ICLR. 2020
> [5] Confident Adaptive Language Modeling. NeurIPS. 2022
> [6] Structured Pruning for Efficient Generative Pre-trained Language Models. ACL findings. 2023
> [7] Block Pruning For Faster Transformers. EMNLP. 2021
> [8] A Fast Post-Training Pruning Framework for Transformers. NeurIPS. 2022
> [9] What Matters In The Structured Pruning of Generative Language Models?. ArXiv. 2023

---

### Meta-Review · Area_Chair_LaqT · 2023-09-17

**Recommendation:** 4

**Metareview:**

The paper explores structured pruning techniques for encoder-decoder transformer-based pre-trained models and uncovers novel insights into pruning these models.

**Pros**:
- The approach is well-grounded, as it begins with preliminary experiments to assess the significance of each module in terms of performance and inference speedup.
- The proposed approach outperforms merely applying previous methods without thoughtful adaptation.
- The experiments conducted are thorough and comprehensive, encompassing a range of tasks, including standard fine-tuning tasks and instruction tuning.
- In summary, I believe this work lays a strong foundation for further investigation into effective pruning approaches for encoder-decoder models.

**Cons**:
- Some reviewers express concerns regarding the novelty of the work, as it primarily involves making straightforward adaptations to prior research like CoFi.

---

### Decision · Program_Chairs · 2023-10-07

**Decision:**

Accept-Findings

**Comment:**

The paper explores structured pruning techniques for encoder-decoder transformer-based pre-trained models and uncovers novel insights into pruning these models.

**Pros**:
- The approach is well-grounded, as it begins with preliminary experiments to assess the significance of each module in terms of performance and inference speedup.
- The proposed approach outperforms merely applying previous methods without thoughtful adaptation.
- The experiments conducted are thorough and comprehensive, encompassing a range of tasks, including standard fine-tuning tasks and instruction tuning.
- In summary, I believe this work lays a strong foundation for further investigation into effective pruning approaches for encoder-decoder models.

**Cons**:
- Some reviewers express concerns regarding the novelty of the work, as it primarily involves making straightforward adaptations to prior research like CoFi.